# A spatial map of hepatic mitochondria uncovers functional heterogeneity shaped by nutrient-sensing signaling

Sun Woo Sophie Kang[1], Rory P. Cunningham[1], Colin B. Miller[1], Lauryn A. Brown[1], Constance M. Cultraro[1], Adam Harned[2,3], Kedar Narayan [2,3], Jonathan Hernandez[4], Lisa M. Jenkins [5], Alexei Lobanov[6], Maggie Cam[6] & Natalie Porat-Shliom [1] ✉

In the liver, mitochondria are exposed to different concentrations of nutrients due to their spatial positioning across the periportal and pericentral axis. How the mitochondria sense and integrate these signals to respond and maintain homeostasis is not known. Here, we combine intravital microscopy, spatial proteomics, and functional assessment to investigate mitochondrial heterogeneity in the context of liver zonation. We find that periportal and pericentral mitochondria are morphologically and functionally distinct; beta-oxidation is elevated in periportal regions, while lipid synthesis is predominant in the pericentral mitochondria. In addition, comparative phosphoproteomics reveals spatially distinct patterns of mitochondrial composition and potential regulation via phosphorylation. Acute pharmacological modulation of nutrient sensing through AMPK and mTOR shifts mitochondrial phenotypes in the periportal and pericentral regions, linking nutrient gradients across the lobule and mitochondrial heterogeneity. This study highlights the role of protein phosphorylation in mitochondrial structure, function, and overall homeostasis in hepatic metabolic zonation. These findings have important implications for liver physiology and disease.

Mitochondria are highly dynamic organelles that play critical roles in cell physiology, including energy production through oxidative phosphorylation (OXPHOS), metabolic signaling pathways, and biosynthesis[1–5]. In response to changes in the environment, such as nutrient availability, mitochondria undergo remodeling through fission, fusion, and mitophagy[2,4]. These dynamic rearrangements in mitochondrial architecture are fundamental to mitochondrial function, morphology, and homeostasis[4,5]. Further, these dynamic changes

are modulated, in part, through post-translational modifications, like phosphorylation, allowing mitochondria to rapidly and reversibly adjust their metabolic output[6–8]. In vivo, mitochondria in cells within organs are exposed to varying levels of nutrients due to their spatial positioning with respect to the blood supply. How this affects and/or regulates mitochondrial dynamics is not known.

This is particularly important in the liver, a central metabolic organ that balances whole-body nutrient availability. Within the liver,

[1]Cell Biology and Imaging Section, Thoracic and GI Malignancies Branch, National Cancer Institute (NCI), National Institutes of Health (NIH), Bethesda, MD, USA. [2]Center for Molecular Microscopy, Center for Cancer Research, National Cancer Institute, National Institutes of Health, Bethesda, MD, USA. [3]Cancer Research Technology Programs, Frederick National Laboratory for Cancer Research, Frederick, MD, USA. [4]Surgical Oncology Program, National Cancer Institute (NCI), National Institutes of Health (NIH), Bethesda, MD, USA. [5]Laboratory of Cell Biology, National Cancer Institute (NCI), National Institutes of Health (NIH), Bethesda, MD, USA. [6]CCR Collaborative Bioinformatics Resource (CCBR) National Cancer Institute (NCI), National Institutes of Health (NIH), Bethesda, MD, USA. ✉e-mail: poratshliomn@mail.nih.gov

hepatocytes are organized into polygonal units called lobules. Nutrient-rich blood flows unidirectionally into the lobule, entering through the hepatic artery and portal vein (which will hence be referred to as periportal; PP) and drains into a single central vein (pericentral; PC). Consequently, the hepatocytes are exposed to different levels of regional metabolites and metabolic burdens depending on their relative location within the lobule[9,10]. It has been previously shown that these gradients direct hepatocytes in different parts of the lobule to express different genes, a phenomenon known as liver zonation[10,11]. Wnt ligands secreted by PC endothelial cells are a major driver of zonal gene expression[12–14]. Although single-cell RNA sequencing has enhanced our understanding of liver zonation[15,16], the functional consequences are often inferred solely from gene expression.

Electron microscopy studies have revealed notable differences in mitochondrial morphology between PP and PC hepatocytes[17–20]. These observations suggest that cells on the PP–PC axis, separated by up to a mere 300 μm, possess mitochondria with distinct functions. However, it is not known how spatial separation affects mitochondria functions in vivo. Furthermore, whether mitochondrial variations are determined genetically or continuously adjusted metabolically via dynamic nutrient gradients requires elucidation.

In the present study, we aimed to define the relationship between mitochondrial function, structure, and spatial positioning in the hepatic lobule. We also sought to identify mechanisms that regulate mitochondrial diversity during homeostasis. Our results reveal hepatic mitochondrial zonation, combining structural and functional features with specialized mitochondrial subpopulations within the liver lobule. Further, they highlight the role of protein phosphorylation and nutrient sensing in dynamically tuning zonated mitochondrial functions.

## Results

### A comparative mitochondrial proteome of sorted hepatocytes

To investigate how the spatial positioning of cells in the liver affects mitochondrial functions, PP and PC hepatocytes from the livers of four ad-lib-fed mice were enriched using unique surface markers and fluorescence-activated cell sorting (FACS; Fig. 1A–C). E-cadherin and CD73 antibodies were used to enrich PP and PC hepatocytes, respectively, and Western blots were performed for validation (Fig. 1D).

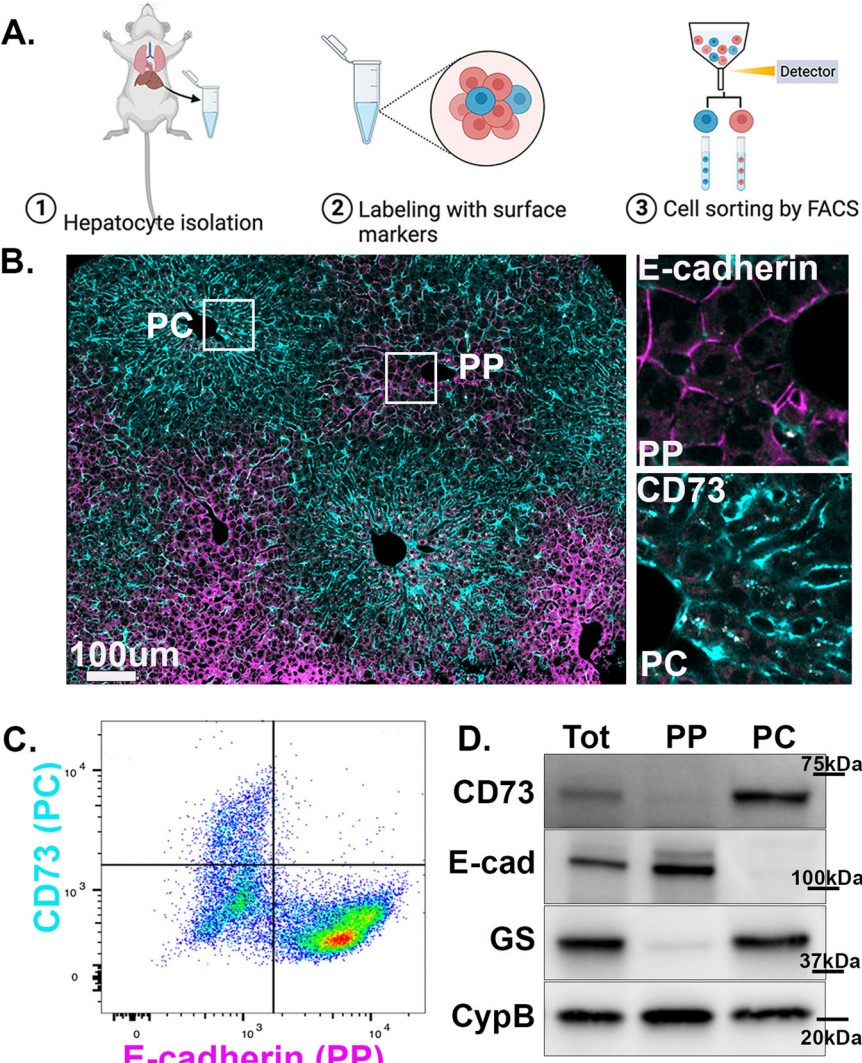

**Fig. 1 | Spatial enrichment of PP and PC hepatocytes. A** Schematic diagram depicting the workflow. Hepatocytes were isolated from the murine liver using two-step collagenase perfusion, after which unique surface markers were applied to label cells. Labeled cells were then enriched using fluorescence-activated cell sorting (FACS) to obtain hepatocytes from different zones. Illustration created using BioRender. **B** Immunofluorescence staining of a liver section showing the zonal distribution of CD73 (cyan) and E-cadherin (magenta). Representative image from three independent experiments. **C** Representative two-dimensional scatter plot of hepatocytes labeled with CD73 and E-cadherin. **D** Western blot of spatially sorted hepatocytes. Representative blot from $n = 3$ independent experiments. Source data is provided as a Source Data file for (**D**). PP periportal, PC pericentral, GS glutamine synthetase.

Next, PP and PC hepatocytes were subjected to tandem mass tag (TMT)-based quantitative mass spectrometry for total proteome analysis. The goal was to gain insight into mitochondrial functions by establishing a quantitative map of mitochondrial protein abundance along the PP–PC axis. Principal component and hierarchical clustering analyses showed sample grouping based on spatial origin (PP or PC) (Supplementary Fig. 1A, B). Of 5018 proteins identified, 46% were zonated, meaning they had a biased expression toward PP or PC hepatocytes (Fig. S1C, D; Supplementary Data 1). Pathway analysis highlighted PP and PC-restricted processes consistent with a previous study describing gene expression[16] (Supplementary Fig. 1E, F).

The list of quantified proteins was compared with the murine MitoCarta 3.0 database, which consists of 1,140 proteins. We identified 829 mitochondrial proteins, 422 of which were enriched in PP mitochondria and 113 in PC mitochondria (Fig. 2A, B). To gain insight into the functions of PP and PC mitochondria, the top 25 unique proteins were selected, and their location and pathway within the mitochondria were determined using MitoCarta 3.0 database (Fig. 2C, D). Selected mitochondrial proteins were also validated by immunofluorescence (Supplementary Fig. 2).

Proteins enriched in PP mitochondria were primarily localized to the inner membrane or the mitochondrial matrix and were involved in amino acid metabolism or OXPHOS. Thus, we examined the spatial expression of selected OXPHOS proteins and found that PP mitochondria expressed higher levels of nuclear and mitochondrial-encoded OXPHOS components (Fig. 2E). In contrast, proteins enriched in PC mitochondria localized to the outer and inner mitochondrial membranes and matrix. In addition to regulating mitochondrial structure, dynamics, and contact with other organelles, many of the identified PC proteins are involved in lipid metabolism, detoxification, and carbohydrate metabolism (Fig. 2D). Notably, citrate synthase (CS), a tricarboxylic acid cycle (TCA cycle) enzyme, was highly expressed in PC mitochondria. In addition to the TCA cycle, when transported into the cytosol, citrate can be a precursor for lipid synthesis. Indeed, STRING analysis suggested a functional link between PC mitochondrial proteins involved in pyruvate metabolism and phospholipid membrane biosynthesis and lipid synthesis occurring in the cytosol (Fig. 2F).

### PP mitochondria display enhanced bioenergetic capacity

We next performed a functional evaluation of the PP and PC mitochondria to determine their bioenergetic capacity. Intravital microscopy[21] was used to examine mitochondrial membrane potential in the intact liver of anesthetized mice. Mitochondria were labeled with MitoTracker green and TMRE, with the former labeling mitochondria matrix and the latter indicating membrane potential. TMRE labeled mitochondria in PP regions more intensely, indicating higher membrane potential (Fig. 3A, B and Supplementary Fig. 3A, B). Isolated hepatocytes were labeled with Anti-CD73, Anti-E-cadherin antibodies, and JC1, a ratiometric fluorescent reporter of mitochondrial membrane potential. Consistent with the in vivo data, PP cells labeled in suspension had higher mitochondrial membrane potential than PC cells (Fig. 3C, D). Together, these data show that PP mitochondria have a higher membrane potential that is not disrupted by spatial sorting, allowing the use of these cells for further physiological evaluation.

Subsequently, mitochondrial oxygen consumption rate and substrate preferences were evaluated using the Seahorse XF Analyzer in spatially sorted hepatocytes. PP hepatocytes consumed up to double the amount of oxygen compared to PC cells (Fig. 3E) and had higher maximal respiration (Fig. 3F). Next, substrate preference was determined by measuring the rate of ATP production in the presence of inhibitors. The capacity of PP mitochondria to produce ATP was significantly decreased by etomoxir, an irreversible inhibitor of fatty acid oxidation, whereas, in PC mitochondria, UK5099, an inhibitor of the mitochondrial pyruvate carrier that inhibits pyruvate-dependent oxygen consumption, negatively impacted ATP production (Fig. 3G, H).

ATP levels in PP hepatocytes were significantly higher as measured by a luminescent ATP Detection Kit (Fig. 3I). On the other hand, citrate synthase expression and activity as well as triglyceride (TG) levels were higher in PC hepatocytes (Fig. 2D and Fig. 3J, K). Likewise, lipid droplets (LDs) which store TGs, were more abundant in PC regions of the lobule (Supplementary Fig. 4A). Higher TG levels in PC cells could be the result of lower lipid oxidation, increased lipid uptake, or decreased lipophagy. Alternatively, since high ATP levels inhibit citrate synthase activity[22,23], the lower ATP levels in PC hepatocytes may permit citrate synthase activity and lipogenesis.

Consistent with a previous report[24], mRNA and protein abundance of the lipogenesis-related enzymes Fasn (fatty acid synthase), Acly (ATP citrate synthase), Acaca (ACC1, acetyl-CoA carboxylase), and Scd1 (Stearoyl CoA desaturase 1) displayed a PP bias or were unzonated (Supplementary Fig. 4C, D). However, PP hepatocytes displayed higher levels of serine 79 phosphorylation, on acetyl-CoA carboxylase (ACC1), which inhibits lipogenesis supporting the hypothesis that lipogenesis preferentially occurs in PC hepatocytes (Supplementary Fig. 4B). This implies that the functional specialization of hepatocytes is not only regulated by differential expression but also via phosphorylation.

### Mitochondrial organization varies across the lobule

To better understand the spatial organization of mitochondria in the hepatic lobule, we used confocal microscopy to visualize mitochondria in liver sections from mito-Dendra2 mice[25]. Consistent with previous reports[17–20], an apparent dichotomy in mitochondrial architecture was observed along the PP–PC axis, with short, round mitochondria in PP and tubulated mitochondria in PC hepatocytes (Fig. 4A and Movies S1 and S2). The transition between these two phenotypes occurred in the mid-lobular area, with a limited number of cells containing both phenotypes within a single cell (Fig. 4A).

Mitochondrial topology was visualized in 3D and at nanometer resolution with Focused Ion Beam Scanning Electron Microscopy (FIB-SEM)[26,27]. Initially, scanning electron microscopy images of the hepatic lobule were used to identify PP and PC regions for FIB-SEM (Supplementary Fig. 5). Next, FIB-SEM volumes of the selected cells were acquired, individual mitochondria segmented, and mitochondrial volumetric models generated (Fig. 4B, C and Movies S3 and S4; see Material and Methods). Distinct subcellular organization of the mitochondrial network in different parts of the lobule was apparent (Fig. 4B, C). Mitochondria volume and surface area in PP cells were approximately 2-fold greater than those measured in PC cells (Fig. 4D, E). The sphericity index also significantly differed between PP and PC, with mean values of 0.93 and 0.74, respectively (Fig. 4F). The higher mean values in PP hepatocytes indicate a shape closer to a sphere as demonstrated by the round surfaces rendered (Fig. 4C).

Larger mitochondrial volume in PP hepatocytes, together with higher bioenergetic capacity (Fig. 3) suggests that overall mitochondrial mass is higher in PP regions of the lobule. Lending further support to this idea, mitochondrial DNA copy number, a commonly used method to evaluate mitochondrial mass, was also higher in PP hepatocytes (Fig. 4G). Mitochondrial structural diversity, including the typical PP and PC morphologies described above, was conserved in the human liver, suggesting a similar structure-function relationship may apply (Supplementary Fig. 6). Taken together, mitochondrial structural variations across the lobule strongly correlate with the functional diversity in lipid oxidation and biosynthesis under normal physiological conditions.

### Enhanced mitophagy flux in PC hepatocytes

Mitophagy, the selective degradation of mitochondria via autophagy, is stimulated in response to various signals, including hypoxia, nutrient deprivation, and glucagon signaling[28,29]. In the liver, basal mitophagy is also activated by the daily feeding and fasting cycle[30,31]. To evaluate if mitophagy is differentially regulated across the liver lobule, we applied

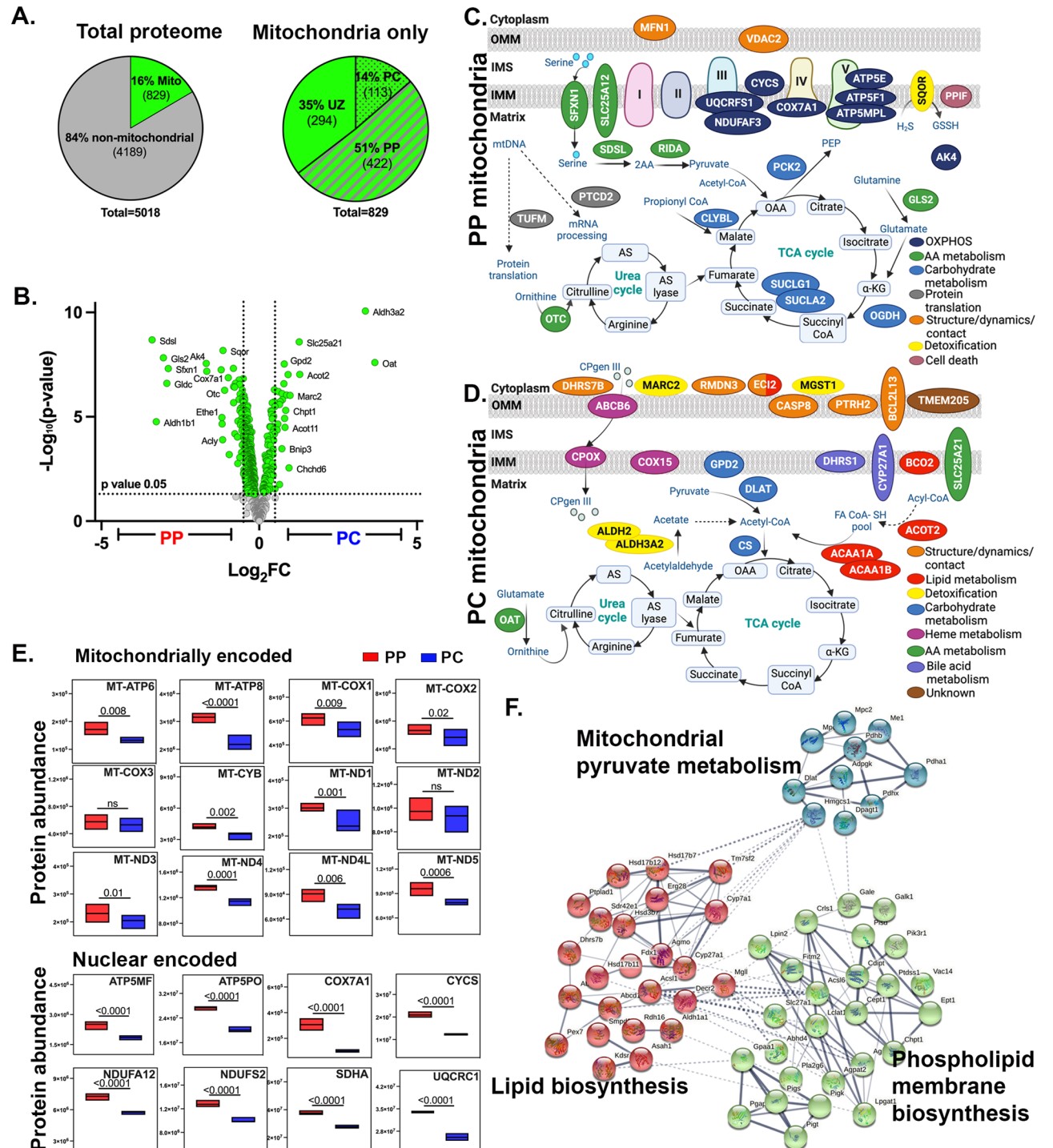

**Fig. 2 | Comparative mitochondrial proteome of spatially sorted hepatocytes. A** Total number of non-mitochondrial proteins (gray) and mitochondrial proteins (green) was detected by mass spectrometry (left; $n = 4$ mice). Percentage of periportal (PP), pericentral (PC), and unzoned (UZ) mitochondrial proteins (right); zonated expression based on a $p$-value (0.05). **B** Volcano plot shows the $\log_2$ PC/PP fold-change (x-axis) and the $-\log_{10}$ $p$-value (y-axis) for mitochondrial proteins. Proteomics data was analyzed with Limma R package (v3.40.6). **C**, **D** Spatial distribution of the top 25 PP or PC mitochondrial proteins. Proteins were color-coded to reflect their cellular

function. Pathways were listed based on the frequency at which they appeared. **E** Abundance of representative nuclear and mitochondria-encoded respiratory chain proteins are shown with floating bar graphs. The center represents the mean, top and bottom represent maxima and minima. The $p$-values were calculated with Limma R package (v3.40.6). **F** Bioinformatic STRING analysis of the PC mitochondria proteomic data. The interaction map illustrates the functional association of PC mitochondrial metabolism with cytosolic lipid synthesis. Data presented as mean ± SD. *$p < 0.05$, **$p < 0.01$, ***$p < 0.001$, ****$p < 0.0001$.

intravital microscopy of mitochondria-targeted Keima in ad-lib-fed mice (mtKeima)[32]. The mtKeima protein has different excitation wavelengths depending on the acidity of the mitochondrial environment; in a neutral environment, mtKeima excitation is at 440 nm, while

in an acidic environment, excitation is at 560 nm (Fig. 5A). The ratio of acidic-to-neutral excitation is a measure of mitophagy and the relative difference between mitophagy in the presence or absence of the protease inhibitor leupeptin, is used to determine mitophagy flux. Tile

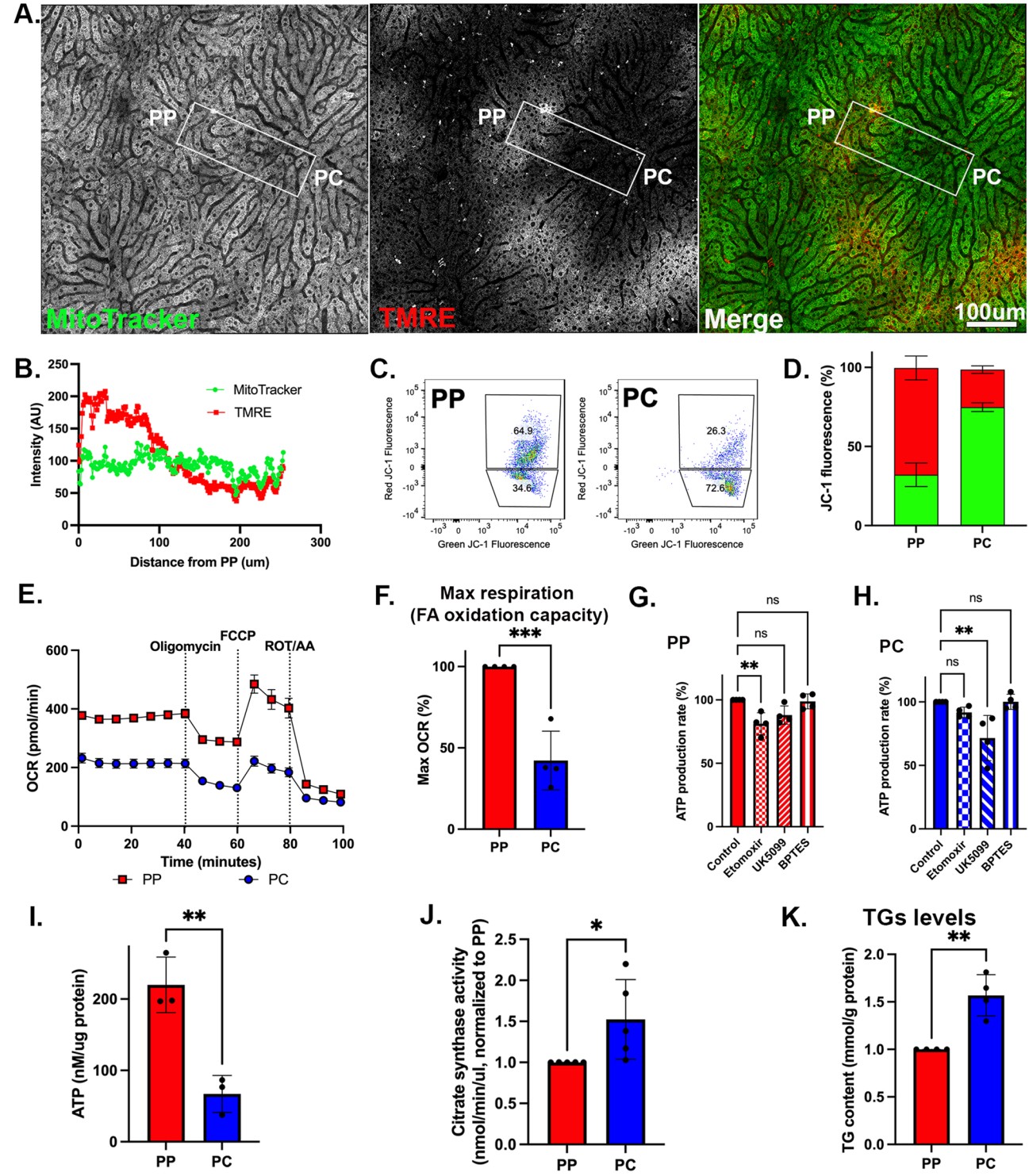

scans of entire lobules were acquired and ratios of acidic-to-neutral excitation were calculated in selected regions (Fig. 5B). While in saline-treated mice, mitophagy was consistently higher in PP regions, leupeptin significantly increased mitophagy only in PC hepatocytes, suggesting a higher mitophagy flux in PC regions. (Fig. 5C). Notably, the increase in mtKeima fluorescence in PP regions was driven by a higher signal in both the neutral and acidic compartments, while in PC regions it was mainly due to an increase in the acidic compartments (compare grayscale insets with and without leupeptin in Fig. 5B). Similar trends were obtained with another mitophagy reporter Cox8-

EGFP-mCherry[33] (Supplementary Fig. 7A, B). The spatial differences in mtKeima fluorescence intensities in PP and PC regions (Fig. 5B) were not due to variations in mtKeima expression levels (Supplementary Fig. 7C).

We further tested the spatial differences in mitophagy by evaluating the levels of the mitophagy receptor Bnip3 and the autophagy marker LC3A/B using Western blots (Fig. 5D). In saline-treated mice, there were higher levels of Bnip3 in PC hepatocytes, and significantly higher levels in mice treated with leupeptin, consistent with higher mitophagy flux in PC regions (Fig. 5D, E). On the other hand, the

**Fig. 3 | PP mitochondria display higher bioenergetic capacity. A** Intravital microscopy of the hepatic lobule labeled with tetramethylrhodamine ethyl ester (TMRE), and MitoTracker Green to evaluate the relative mitochondrial membrane potential. The scale bar is 100 μm. Similar trends were observed in three independent experiments. **B** Mitochondrial membrane potential was evaluated by measuring fluorescence intensity (AU) along a PP–PC axis line. **C, D** Measurement of mitochondrial membrane potential using JC1 and flow cytometry in spatially sorted hepatocytes. Dot plots of a representative experiment is shown. The bar graph shows similar trends in three independent experiments. Stacked bar plot of relative percentages of green or red positive cells from $n = 3$. Two-way ANOVA, Šidák multiple comparisons test $p < 0.0001$. **E** Oxygen consumption rate (OCR) in spatially sorted hepatocytes using the XF Mito Stress Test Kit and Seahorse XF96 Analyzer. Samples were normalized to cell number. Similar trends were observed in four independent experiments. **F** Maximum respiration capacity in spatially sorted hepatocytes expressed relative to PP. Data presented as mean ± SD from $n = 4$

independent experiments. Statistical significance was calculated with two-tailed unpaired Student's $t$-test ***$p = 0.0007$. **G, H** Substrate dependency assay in spatially sorted hepatocytes using the MitoFuel Flex test. ATP production rate relative to PP is shown in cells treated with etomoxir, UK5099, or BPTES. Data presented as mean ± SD from $n = 4$ independent experiments. Statistical significance was calculated using one-way ANOVA, Dunnett's multiple comparisons test; ns not significant; **$p = 0.003$. **I** ATP content relative to PP in spatially sorted hepatocytes using the colorimetric luciferase assay from three independent experiments. **J** Citrate synthase activity relative to PP in spatially sorted hepatocytes. Data presented as mean ± SD from $n = 5$ independent experiments. Statistical significance was calculated using two-tailed unpaired Student's $t$-test *$p = 0.04$. **K** Intracellular triglyceride (TG) concentration relative to PP in spatially sorted hepatocytes. Data presented as mean ± SD from $n = 4$ independent experiments. Statistical significance was calculated using two-tailed unpaired Student's $t$-test **$p = 0.002$. Periportal (PP) is labeled in red; pericentral (PC) in blue.

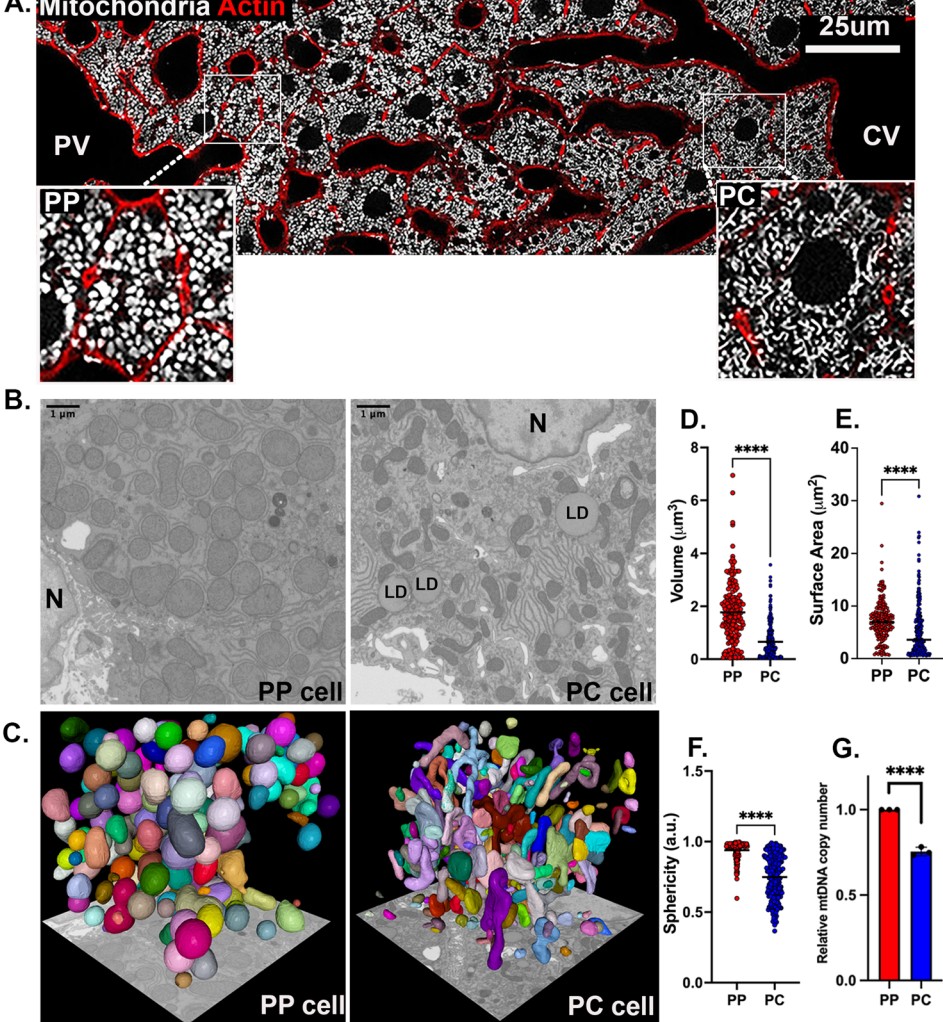

**Fig. 4 | Mitochondrial morphology and organization across the lobule.**
**A** Confocal image of the PP–PC axis in liver sections from Mito-Dendra2 transgenic mice. Mitochondria are shown in white, and actin was labeled with phalloidin in red. Enlarged insets of a representative PP (left) and PC (right) cell are shown.
**B** Mitochondria were visualized by Focused Ion Beam Scanning Electron Microscopy (FIB-SEM). Representative sections of PP (left) and PC (right) cells. N Nucleus, LD Lipid droplet. **C** Segmentation and volume rendering of mitochondria from PP (left) and PC (right) cells using MitoNet and empanada-napari. **D–F** Quantification

of mitochondrial morphological features including volume, surface area, and sphericity index (a measure of similarity to a perfect sphere (=1)). A collection of 175 mitochondria in PP and 250 in PC were analyzed and statistical significance calculated by two-sided Mann–Whitney test. Data presented as mean ± SD; ****$p < 0.0001$. **G** Quantification of relative mtDNA copy number by qPCR in spatially sorted hepatocytes. The bar graph shows four independent experiments. Data presented as mean ± SD; ****$p < 0.0001$. Periportal (PP) is labeled in red; pericentral (PC) in blue.

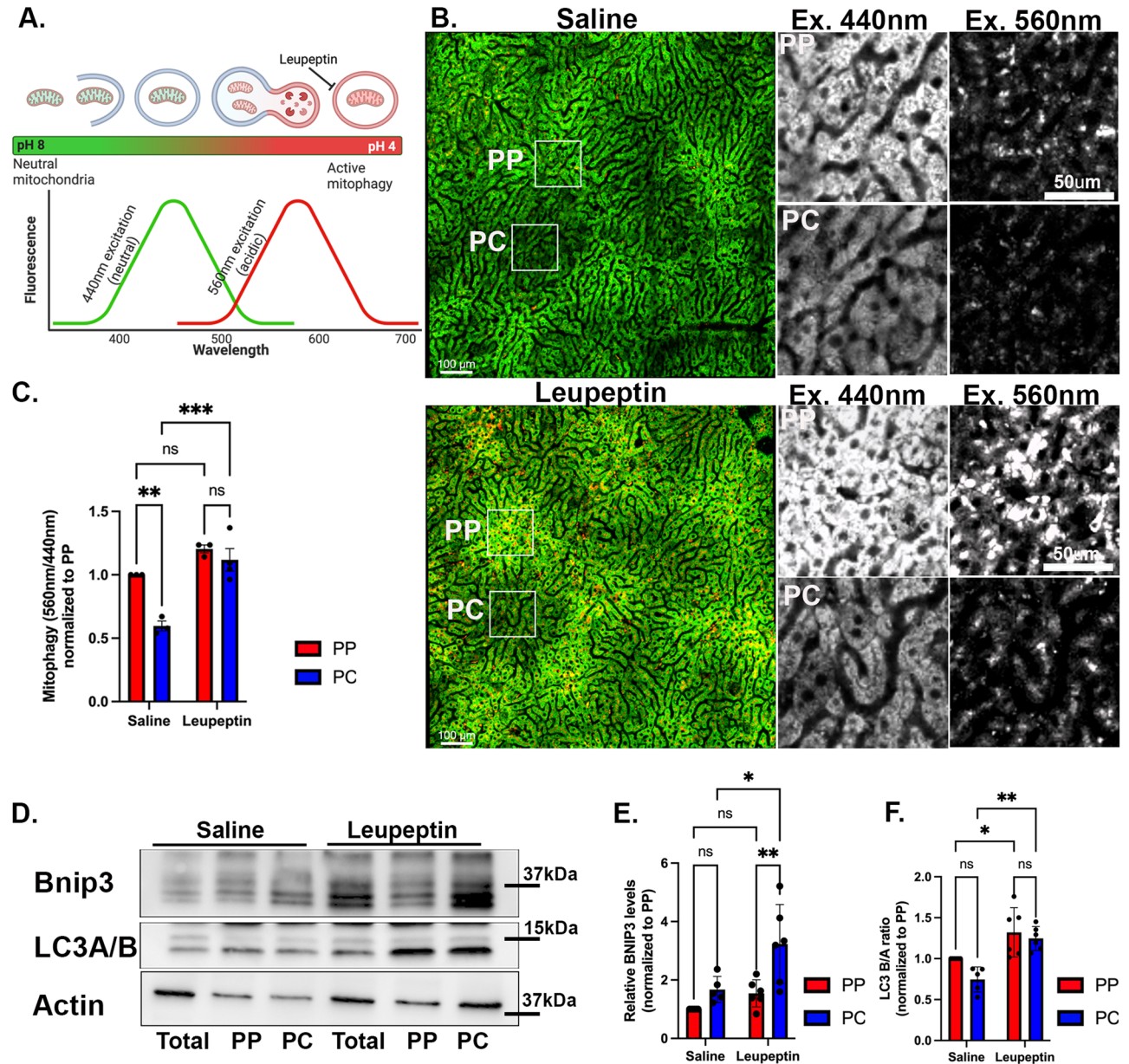

**Fig. 5 | PC mitochondria display higher turnover via mitophagy. A** mtKeima is a pH-sensitive mitophagy reporter. The ratio between the 440 nm (green; neutral pH) and 560 nm (red; acidic pH) excitation measures the proportion of mitochondria undergoing mitophagy. Illustration created using BioRender. **B** Intravital microscopy of transgenic mice expressing mtKeima in saline-injected or leupeptin-injected mice. Representative images of the hepatic lobule and magnified insets of PP and PC regions. **C** Quantification of mitophagy in PP and PC regions expressed as a fold-change relative to PP cells. Data presented as mean ± SD from *n* = 3 independent experiments. Statistical significance was

calculated using two-way ANOVA, Tukey's multiple comparisons test; ns not significant; **p* = 0.03, ***p* = 0.001. **D** Immunoblots of Bnip3 and LC3A/B of unsorted (total) and sorted PP and PC hepatocyte populations from livers treated with either saline or leupeptin. Mouse 1 (M1); Mouse 2 (M2). **E–F** Quantification of Bnip3 expression and LC3B/A ratio. Data presented as mean ± SD from *n* = 5 (saline) or *n* = 6 (leupeptin) independent experiments. Statistical significance was calculated using two-way ANOVA, Tukey's multiple comparisons tests; ns not significant; **p* = 0.02 and ***p* = 0.006 (Bnip3) ;**p* = 0.03 and ***p* = 0.0014 (LC3B/A). Periportal (PP) is labeled in red; pericentral (PC) in blue.

proportion of LC3A/B (an autophagy marker) was similar in PP and PC hepatocytes, and leupeptin treatment had a comparable effect on both PP and PC cells, suggesting uniform levels of basal autophagy across the lobule (Fig. 5D, F). Other mitophagy-related proteins were enriched in PC hepatocytes (Supplementary Fig. 7D), as previously reported[34,35]. Likewise, the lysosomal marker LAMP1 was distributed uniformly across the lobule, while LAMP1-positive lysosomes containing mitochondria were more abundant in PC hepatocytes (Supplementary Fig. 7E). Thus, basal mitophagy is higher in PC hepatocytes, which may be driven by the higher expression of mitophagy-related proteins and contribute to the lower mitochondrial mass.

**Nutrients drive mitochondrial diversity via phosphorylation**

Reversible protein phosphorylation has been linked to cellular and mitochondrial metabolism[6–8,36,37]. To examine the role of phosphorylation in regulating mitochondrial spatial heterogeneity and identify signaling pathways that may be involved, we profiled the phosphoproteomes of PP and PC hepatocytes (Supplementary Fig. 8A–C). We identified 7278 phosphopeptides corresponding to 3686 phosphosites in 1623 proteins (Fig. 6A; Supplementary Data 2). The majority of phosphorylation sites were on serine residues (89%), with a smaller proportion identified on threonine (10%) and tyrosine (1%) (Fig. 6B). More than half of the proteins had a single phosphorylation site

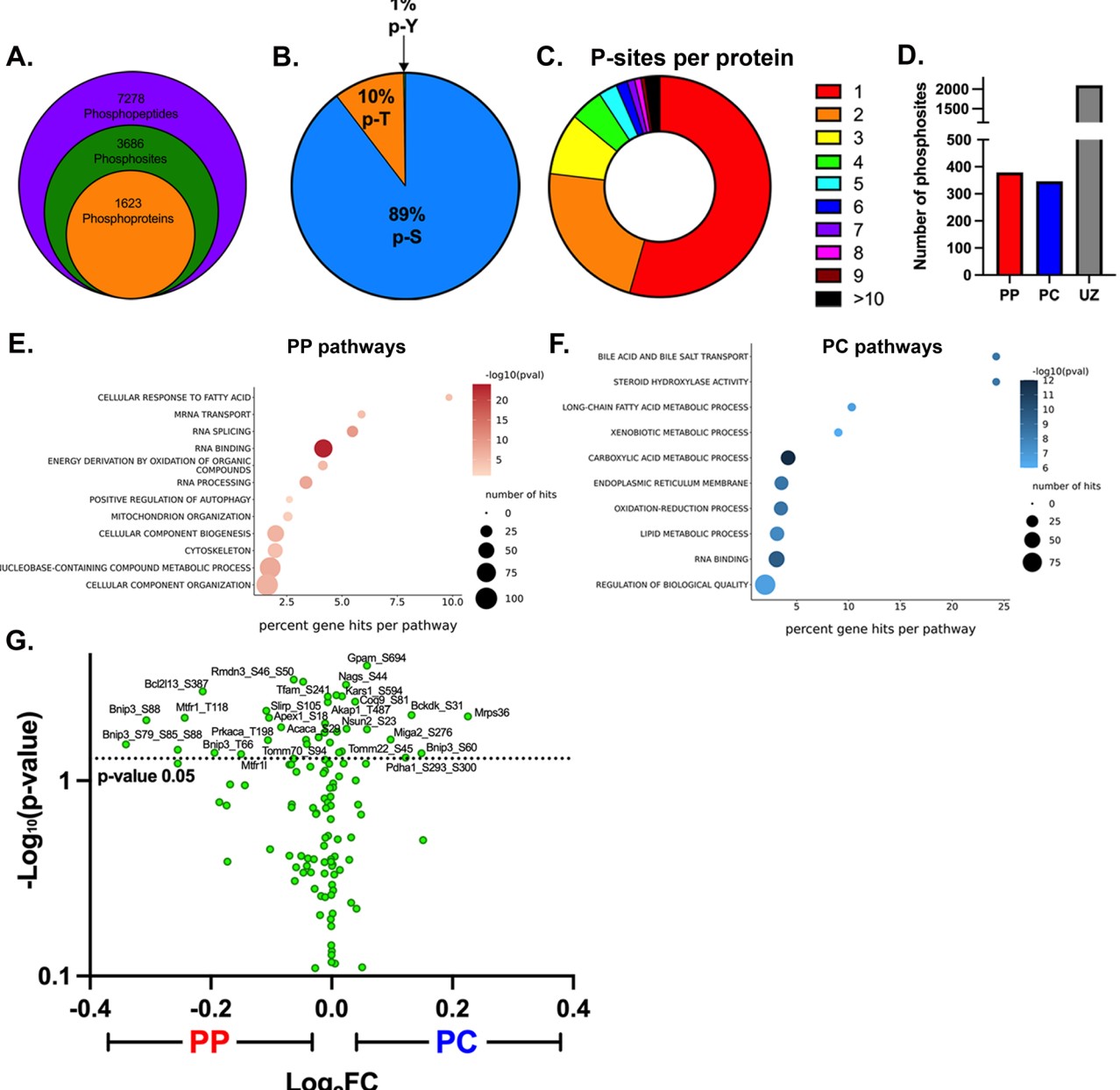

**Fig. 6 | Phosphoproteome highlights spatially regulated pathways contributing to mitochondrial phenotypes. A** Overview of phosphopeptides, phosphosites, and phosphoproteins identified in the phosphoproteome. **B** Proportion of phosphorylated serine (p-S), threonine (p-T), and tyrosine (p-Y) residues in the identified phosphopeptides. **C** Proportion of proteins identified with a certain number of phosphorylated residues per protein. **D** Number of phosphosites with a PP, PC, or unzonated (UZ) bias is shown in the bar graph. **E–F** The PP or PC zonated phosphoproteins were analyzed using GO enrichment analysis. **G** Mitochondrial phosphoproteome is shown in a volcano plot with $\log_2$ PC/PP fold-change *(x-axis)* and the $-\log_{10}$ *p*-value (y-axis).

(Fig. 6C). Interestingly, the overall number of phosphosites with PP or PC zonation was similar (Fig. 6D), suggesting that ATP availability across the lobule (Fig. 3A) is not limiting in homeostasis.

Next, we performed a pathway analysis on zonated phosphopeptides to determine the processes that may be influenced by phosphorylation (either activating or inhibiting) in each hepatocyte population (Supplementary Data 3). In PP hepatocytes, protein translation, metabolism, and positive regulation of autophagy were most notably regulated via phosphorylation (Fig. 6E). On the other hand, multiple pathways related to lipid and phospholipid synthesis were represented in PC cells (Fig. 6F). Sequence analysis of phosphorylated peptides using the online tool Momo (https://meme-suite.org/meme/tools/momo) revealed several phosphorylation motifs were enriched

in different parts of the lobule, suggesting potential involvement of distinct kinases and signaling pathways (Supplementary Table 1).

We also examined the phosphorylation of mitochondrial proteins, with mitochondrial phosphoproteome shown in the volcano plot (Fig. 6G) and the zonated phosphoproteins for each hepatocyte group summarized in the table (Table 1 and Supplementary Table 2). Assessment of upstream regulators/kinases using PhosphoSitePlus highlighted factors such as leptin, insulin, mTOR, and AMPK as major drivers of the zonated phosphorylation (Supplementary Data 4). To further examine the role of nutrient sensing in shaping the zonated phosphoproteome, we surveyed all zonated phosphopeptides for putative AMPK and mTOR consensus sites using Group-based Prediction System (GPS) 5.0. Approximately 50% of the PP or PC

**Table 1 | Zonated mitochondrial phosphoproteins, phospho-sites, and cellular function**

| Protein | P-Zonation | P-site | Function |
|---|---|---|---|
| Acaca | PP | S29[a] | Lipogenesis |
| Akap1 | PP | S101/S103[b]/<br>S104/S109 | cAMP-PKA signaling |
| Apex1 | PP | S18 | mtDNA maintenance |
| Bcl2l13 | PP | S387 | Mitophagy |
| Bnip3 | PP | S79, S85, S88[b], T66[b] | Mitophagy |
| Comt | PP | S261 | Catechol metabolism |
| Ehhadh | PP | T543 | Fatty acid oxidation |
| Gpam | PP | S687, S694 | Phospholipid metabolism |
| Mtfr1 | PP | T118 | Fission |
| Mtfr1l | PP | S234/S235[b] | Fission |
| Nadk2 | PP | S373 | Phosphorylation of NAD |
| Prkaca | PP | T198[b] | cAMP-PKA signaling |
| Rmdn3 | PP | S46[b], S50 | Mito dynamics/contact sites |
| Slirp | PP | S105 | mtRNA stability |
| Tfam | PP | S241 | mtDNA maintenance |
| Tomm70 | PP | S94[a] | Protein import and sorting |
| Akap1 | PC | T487 | cAMP-PKA signaling |
| Bckdk | PC | S31[b] | Amino acid metabolism |
| Bnip3 | PC | S60[b] | Mitophagy |
| Coq9 | PC | S81 | Coenzyme Q metabolism |
| Gpam | PC | S694 | Phospholipid metabolism |
| Hsd17b8 | PC | S58 | Lipid biosynthesis |
| Kars1 | PC | S594 | Protein translation |
| Miga2 | PC | S276 | Fusion/contact sites |
| Mrps36 | PC | S55/T59/S60 | Mitochondrial translation |
| Mtif2 | PC | S180 | Protein translation |
| Nags | PC | S44 | Amino acid metabolism |
| Nsun2 | PC | S23 | RNA modification |
| Pdha1 | PC | S293[a], S300[a] | Pyruvate metabolism |
| Tomm22 | PC | S45 | Protein import and sorting |

Dash and comma distinguish between potential and identified phosphosites, respectively.
*PP* periportal, *PC* pericentral.
[a]Inhibitory.
[b]Activating phosphosites.

phosphorylation sites were identified as putative AMPK or mTOR consensus sequences (Supplementary Fig. 8D). However, there was very little overlap in AMPK or mTOR substrates within PP and PC cells (Supplementary Fig. 8E). This suggests that although active throughout the lobule, AMPK and mTOR act on different substrates in PP and PC hepatocytes. To further substantiate this, we examined the phosphorylation status of two well-characterized mTOR substrates. Whereas phosphorylation on T389 of Ribosomal protein S6 kinase (S6K) displayed a PC bias, phosphorylation on T37/T46 of Eukaryotic translation initiation factor 4E (eIF4E)-binding protein 1 (4E-BP1), showed a PP bias (Supplementary Fig. 8F). Together, these results highlight a potential role for nutrient-sensing signaling in shaping zonated phosphoproteomes.

### Nutrient-sensing signaling shapes mitochondrial diversity

Given the spatial variation in mitochondrial phosphosites, their potential regulation by the nutritional state, and the capacity of mitochondria to respond to nutrient fluctuations[1,8], we hypothesized that nutrient-sensing signaling governs mitochondrial remodeling in hepatocytes. To test this, and decouple protein phosphorylation from transcription, we pharmacologically modulated AMPK or mTOR

signaling in vivo and evaluated mitochondrial membrane potential (JC1), and lipid content (BODIPY) via flow cytometry (Fig. 7A, B). Drug concentrations for acute response were determined by Western blotting of known downstream effectors (Supplementary Fig. 9A, B). Activation of AMPK (AICAR) and the inhibition of mTOR (Torin) were each associated with increased mitochondrial membrane potential in both PP and PC hepatocytes (Fig. 7A). The inhibition of AMPK (Compound C; Cpc) and activation of mTOR (MHY1485) had no impact on membrane potential, suggesting the potential involvement of other factors (Fig. 7A). On the other hand, the activation of mTOR, significantly increased lipid content in both PP and PC cells, highlighting the spatial coordination of these pathways in lipid homeostasis (Fig. 7B). Notably, the drugs had an acute effect on the mitochondria, largely independent of cell size underscoring the complex spatial regulation of nutrient-sensitive signaling in the intact liver (Supplementary Fig. 9C).

Finally, we performed microscopy studies in liver sections to examine if acute modulation of AMPK or mTOR impacted mitochondrial morphology. PC mitochondria from AICAR-treated mice had a round morphology with a higher sphericity index, similar to PP mitochondria in vehicle-treated cells (Fig. 7C, D). Conversely, mTOR activation induced dense, elongated mitochondria in PP hepatocytes, with a lower sphericity index resembling the phenotypes observed in PC hepatocytes (Fig. 7C, D). These effects were observed in multiple cells in the PP and PC regions as shown in z-stacks (Supplementary Fig. 9D). Taken together, the results show that mitochondrial zonation is acutely remodeled in vivo by nutrient-sensing signaling suggesting the nutrient gradient fine-tune mitochondrial metabolic output.

Mitochondrial heterogeneity can be established through various mechanisms, including differential gene expression (i.e. developmentally), dynamic gradients (i.e. metabolically), or a combination. Correlation between Wnt-regulated genes and the mitochondrial proteomes of both PP and PC were examined. We found a moderate negative correlation with the PP mitochondrial proteome and a moderate positive correlation with the PC mitochondrial proteome (Supplementary Fig. 9D). Overall, these experiments show that acute modulation of nutrient-sensing pathways shifts mitochondrial functions to impact mitochondrial functional diversity (Fig. 8). These pathways work in concert with Wnt/β-catenin-regulated genes in determining the mitochondrial proteomes of both PP and PC.

### Discussion

In this study, we performed an in-depth characterization of mouse hepatic mitochondria along the PP–PC axis and assessed the spatial heterogeneity of these mitochondria in two populations of cells (Fig. 1). We showed that mitochondria residing in hepatocytes adjacent to the portal vein produce significantly higher levels of ATP through lipid oxidation and OXPHOS. Roughly 300 μm apart, PC mitochondria oxidize pyruvate and produce higher citrate levels for lipogenesis. This PP–PC dichotomy in mitochondrial function mirrors the spatial separation of lipid utilization and biosynthesis in the liver. Our study provides a spatial assessment of the molecular makeup and functional specialization of hepatic mitochondria. It also provides an in-depth investigation of how the physiological niche within the liver lobule alters mitochondrial metabolism. This location-dependent functional heterogeneity highlights the importance of considering mitochondrial disparity in understanding liver physiology and disease.

Consistent with previous observations[17-20], we also found striking variations in mitochondrial morphology across the PP–PC axis (Fig. 4). Specifically, large spherical mitochondria correlated with enhanced OXPHOS in PP hepatocytes (Figs. 2E and 4A–F). This morphology was previously shown to improve oxygen and nutrient exchange and tightly control calcium concentrations, an essential regulator of OXPHOS and mitophagy[38,39]. Conversely, longer, tubulated

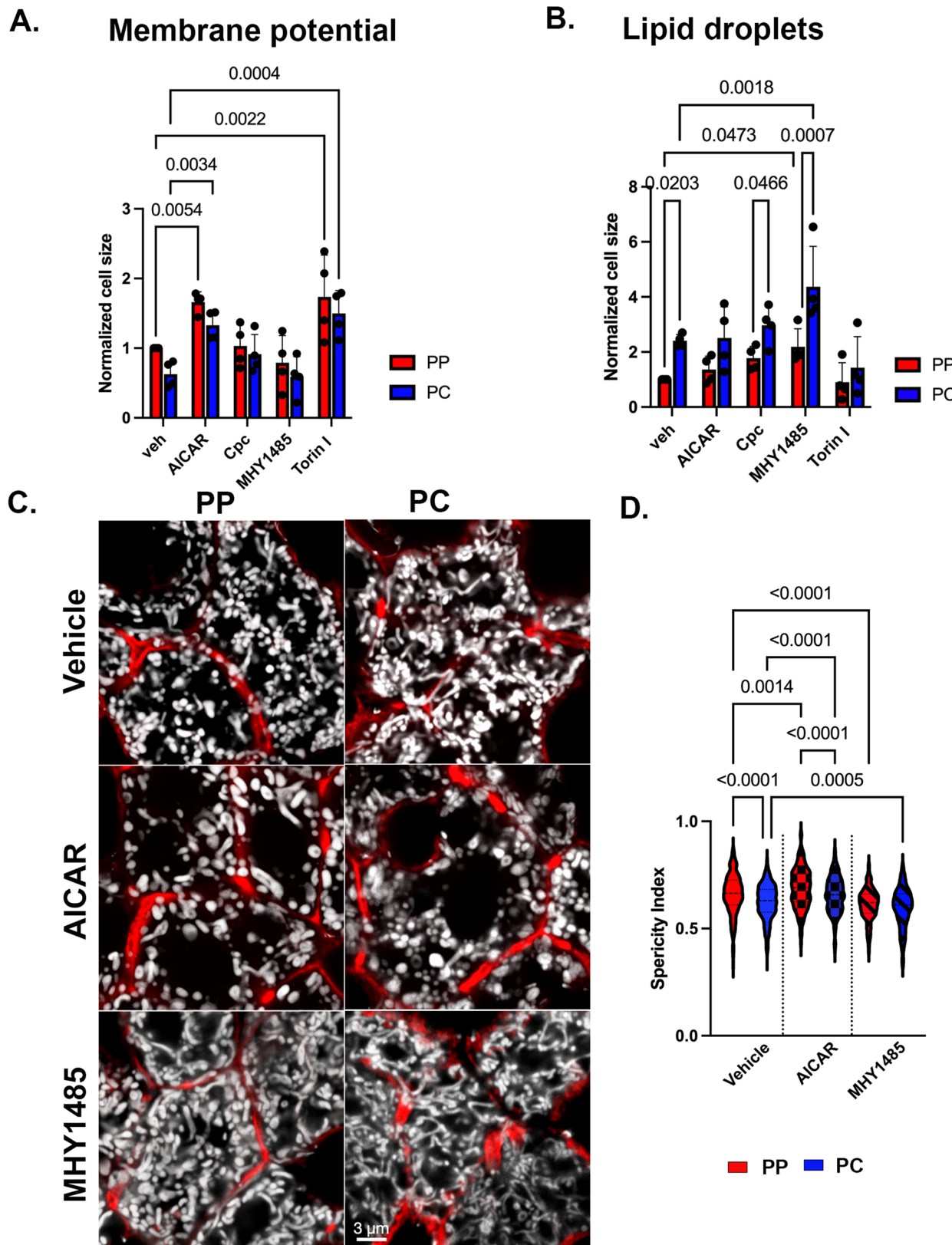

mitochondria (Fig. 4A–F) may protect against mitochondrial degradation[38], counterbalancing the higher mitophagy flux in PC regions (Fig. 5A–C). Mitochondrial tubular versus spherical morphology may also impact mitochondrial metabolic tasks; with tubular mitochondria favoring enzymatic activities in the matrix, and spherical mitochondria improving bioenergetic efficiency. Ngo and colleagues recently demonstrated that mitochondrial fragmentation, such as observed in PP hepatocytes, increases fatty acid oxidation[40]. We further demonstrated a correlation between hepatic mitochondrial structure and function; pharmacological modulation of major nutrient signaling pathways resulted in a remodeled mitochondrial structure and its corresponding function (Figs. 7 and 8).

**Fig. 7 | Nutrient-sensing signaling contributes to mitochondrial functional and morphological diversity. A**, **B** AMPK or mTOR signaling was modulated in vivo by injecting an activating drug, AICAR or MHY1485, or an inhibitory drug Cpc or Torin, respectively. The impact on mitochondrial membrane potential (JC1) and lipid droplets (BODIPY) was evaluated using flow cytometry. Data presented as mean ± SD from $n = 4$ independent experiments. Statistical significance was calculated using two-way ANOVA, and Fishers Least Significant Difference (LSD) test.

**C** Confocal images of hepatocytes from liver sections of Mito-Dendra2 mice treated with vehicle, AICAR, or MHY1485. Mitochondria are shown in white, and phalloidin outlines hepatocytes in red. Scale bar = 3 μm. **D** Mitochondrial sphericity in mice treated with vehicle, AICAR, or MHY1485 was quantified. Data presented as mean ± SD from $n = 4$ independent experiments. Statistical significance was calculated using two-way ANOVA, and Tukey's multiple comparisons test. Periportal (PP) is labeled in red; pericentral (PC) in blue.

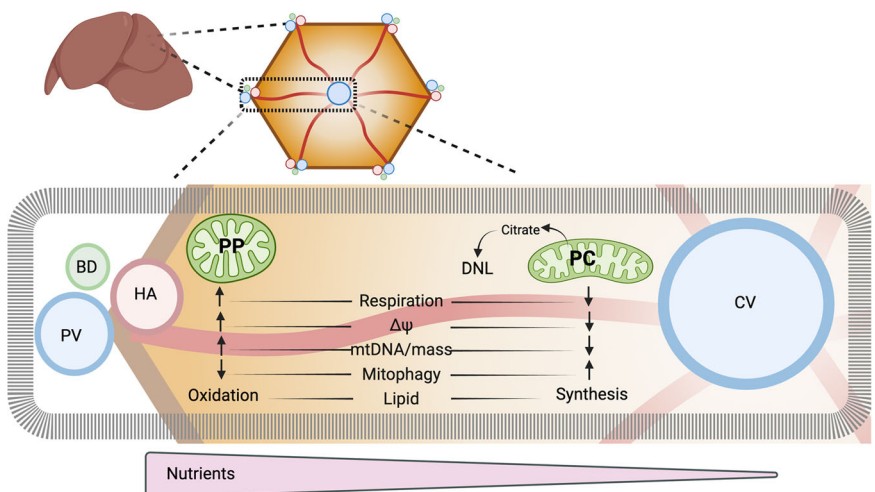

**Fig. 8 | Mitochondrial zonation in the hepatic lobule.** Mitochondria exhibit diversity in both topology and functions along the PP–PC axis. This variation is influenced by nutrient gradients, nutrient-sensing signaling, and subsequent phosphorylation processes. These factors enable hepatic mitochondria to promptly and reversibly adapt to fluctuations in nutrient supply.

Variations in mitochondrial fission and fusion are also likely to contribute to structural diversity, as has been previously shown in other tissues[41,42]. We identified several candidates that may be involved in fusion pathways by regulating mitochondrial dynamics and contact sites in PC mitochondria (Fig. 2D). On the other hand, mitochondrial morphology in PP hepatocytes suggests that fission is favored. Mitochondrial fission can be mediated by the phosphorylation of Mtfr1 or Mtfr1l, both of which showed a PP bias in our phosphoproteomic analysis (Table 1 and Supplementary Table 2). Indeed, phosphorylation of the same conserved site in human Mtfr1l by AMPK was shown to cause mitochondrial fragmentation[43]. Further experiments are necessary to examine the spatiotemporal regulation of mitochondrial fission-fusion dynamics. Our finding that mitochondrial morphology is conserved in the human liver (Supplementary Fig. 6), suggests that some mechanisms are preserved across species and warrant further investigation.

Hepatic mitochondria not only display structural and functional heterogeneity, but they also possess distinct phosphoproteomes associated with each mitochondrial population. Despite higher ATP levels in PP hepatocytes (Fig. 3I), the number of phosphosites was similar to in PC (Fig. 6D). This suggests unique cytosolic or mitochondrial kinases or phosphatases are active in different cells. Indeed, we identified spatially distinct kinases reported to regulate the mitochondrial phosphoproteome[6], including Prakaca (PKA catalytic subunit) in PP mitochondria and Bckdk, in PC mitochondria (Table 1 and Supplementary Table 2). Post-translational control of mitochondrial metabolism through phosphorylation is critical across mouse strains, age, fasting/refeeding, and the onset of Type 2 diabetes[36]. Unexpectedly, the inhibitory phosphorylation of Pdha1 was higher pericentrally (Table 1), which seemingly contradicts the Seahorse data showing that PC hepatocytes rely on pyruvate for ATP production (Fig. 3H). However, the relative abundance of Pdha1 is also higher in PC hepatocytes (Supplementary Data 1) which could lead to overall higher levels of the active enzyme. Another example is the different substrates activated by mTOR. Although 4E-BP inhibition and S6K activation are both downstream of mTORC1 activation, and both promote protein synthesis, previous studies suggested that S6K and 4E-BP differentially control cell growth and proliferation. S6K controls cell size but not cell cycle progression[44] whereas 4E-BP controls cell proliferation but not cell size[45]. PP hepatocytes produce albumin and clotting factors posing a high energetic demand for serum protein synthesis[16]. This is consistent with highly bioenergetic mitochondria and higher phosphorylation of 4E-BP1 that allows protein synthesis. On the other hand, PC hepatocytes are larger (Supplementary Fig. 9C) which is consistent with higher levels of pS6K1. Our study adds to this complexity by demonstrating that differential regulation observed across the lobule is fundamental to hepatic cellular homeostasis. While these phosphorylation sites were not characterized in this study, combining the phosphoproteome data with functional studies allowed us to gain a system-level perspective into the functional consequences of protein phosphorylation (Fig. 6).

Our research indicates that protein phosphorylation, in conjugation with zonated protein expression, plays a pivotal role in regulating various aspects of mitochondrial functionality, quality control, and lipid handling in distinct hepatic zones. The primary orchestrator of liver zonation at the transcriptional level is the Wnt/β-catenin signaling pathway[13,14,46]. In agreement, we observed a correlation between the spatial distribution of mitochondrial proteins and genes regulated by the Wnt pathway (Supplementary Fig. 9D). However, we also found that mitochondrial proteins that display zonated phosphorylation, (Table 1 and Supplementary Table 2), are intricately linked with spatially controlled pathways, including lipogenesis (Fig. 3 and Supplementary Fig. 4), mitophagy (Fig. 5 and Supplementary Fig. 7), and mitochondrial OXPHOS (Fig. 3). Based on these observations, we hypothesize that the establishment of liver zonation is not solely dictated by gene expression; instead, it can be dynamically remodeled,

through nutrient signaling pathways and protein phosphorylation (Fig. 8).

The regulation of mitophagy and lipogenesis through reversible phosphorylation is ideal. It can be remodeled by nutrient-sensitive kinases (Fig. 7), thus providing a quick mechanism for mitochondria to adjust their metabolic output. Several phosphorylation sites on the mitophagy receptor Bnip3 identified here are known to promote mitophagy[47–49]. We also demonstrate functional zonation of lipogenesis through periportal inhibition of ACC1 (Supplementary Figs. 4B and 6H). Although additional investigation into the complex spatial coordination of AMPK and mTOR is required, our study shows that these kinases operate on distinct substrates in PP and PC cells (Supplementary Fig. 8D, E), which has a direct impact on mitochondrial phenotypes. Disruption to AMPK signaling via the deletion of Liver Kinase B1 (LKB1), led to the upregulation of PP genes and severe whole-body wasting phenotype reinforcing a link between nutrient sensing and liver zonation[50]. The ability to sense and adapt to metabolic state is especially critical in the liver, given its role in maintaining whole-body glucose homeostasis. We speculate that post-translational modifications provide the flexibility required for these vital metabolic adjustments, compared to the relatively sustained process of gene expression. Although further experiments to test this hypothesis are needed, we predict that protein phosphorylation may be the critical link between the observed discrepancies in function versus protein or gene expression. This finding has significant implications beyond the liver for studies where function is inferred from expression.

Advances in omics approaches and microscopy technologies have opened up new avenues for research and discoveries in hepatic cell biology[10,11]. Although liver zonation was described nearly a century ago, we are only now starting to unravel how liver anatomy and the gradient of factors within it affect liver function at the tissue level. Our study demonstrates the delicate interplay between the function, shape, and space of hepatic mitochondria. We reveal that protein phosphorylation is an additional regulatory layer, offering spatial and temporal control of hepatic activities. Other post-translational modifications are also likely at play. In the current study, due to the sample size required for phosphoproteome and functional assays, we were limited to analyzing only two groups of hepatocytes. Therefore, significant technological improvements are required to allow for further and deeper exploration of liver zonation. Such advances are already underway with single-cell proteomics[51], allowing increased spatial resolution in murine models and human samples. Ultimately, understanding how the local physiological niche affects mitochondrial metabolism may be relevant for the therapeutic modulation of these pathways in liver-related pathologies.

## Methods

### Animal experiments
Experiments were approved by the Institutional Animal Care and Use Committee of the National Cancer Institute and comply with the Guide for the Care and Use of Laboratory Animals (National Institutes of Health publication 86–23, revised 1985). All experiments were conducted during the light cycle on ad libitum-fed (NIH-31 Open Formula, Envigo), eight-to-ten-week-old male mice; C57BL/6J (strain# 000664), Mito-Dendra2 (strain# 018397)[25] obtained from Jackson Laboratories and the mtKeima line (Igs2tm1(CAG-mt-Keima)Fink) were a gift from Dr. Toren Finkel[32]. Sex variations was not considered in the current study.

### Intra-cardiac fixation, tissue processing, and immunofluorescence
Mice were anesthetized with 250 mg/kg Xylazine, and 50 mg/kg Ketamine (diluted in saline) injected intraperitoneally (i.p.). The liver was fixed by transcardial perfusion of ice-cold PBS for 2 min followed by ice-cold 4% paraformaldehyde (PFA) in PBS at a rate of 5 ml/min. Livers were harvested and stored in 4% PFA in PBS overnight and processed in a sucrose gradient before embedding in OCT (Tissue-Tek). Blocks were kept at −80 °C until 10 μm thick slices were made with a cryostat and slides prepared. Slides were stored at −80 °C until thawed, rehydrated, and blocked with 0.1% Triton X-100 and 10% FBS in PBS for 1 h at room temperature. Next, slides were incubated with primary antibody at 4 °C overnight. The following day, slides were washed three times, 15 min each, then incubated with a secondary antibody for 1 h at room temperature. After three 15-min washes, slides were mounted with Fluoromount-G and a coverslip (#1).

### Confocal microscopy and image processing
Tile scans of Mito-Dendra2 (strain# 018397) liver sections were acquired using a Leica SP8 inverted confocal laser scanning microscope using a 63x oil objective and a 1.4 numerical aperture. Images were deconvolved using the LIGHTNING module in LAS X.

### Human tissue collection for immunofluorescence and imaging
Human tissue was obtained with informed consent under an NIH IRB-approved protocol (13-C-0076) for risk-reducing surgery performed on patients with germline genetic mutation(s). All tissues procured, which included liver samples used in this study, were grossly normal as determined by the surgeon and histopathologically normal as determined by a board-certified pathologist. Of note, all tissue was obtained within 20 min of incision. Tissues were fixed overnight in 4% PFA in PBS overnight and processed in a sucrose gradient before embedding in OCT (Tissue-Tek).

### Measurement of mitochondrial membrane potential via intravital microscopy
Mitotracker Green FM (250 μM) and tetramethylrhodamine ethyl ester (TMRE, 200 μM) were consecutively injected retro-orbitally to label mitochondria with/without mitochondrial membrane potential in livers of ad libitum-fed, eight-to-ten-week-old male mice; C57BL/6J (strain# 000664). After 30 min of incubation per dye, the liver was exposed to the microscope stage with the mouse under anesthesia, and images were acquired using excitation at 480 nm and 532 nm. The fluorescence intensity of TMRE and Mitotracker Green FM was analyzed across the PP–PC axis through line scanning with ImageJ. Additionally, Alexa Fluor® 647 anti-mouse/human CD324 (E-cadherin, 0.5 μg/g) was retro-orbitally injected to label PP regions in the liver of Mito-Dendra2 mice (strain# 018397)[52]. Mito-Dendra2 mice were used instead of using Mitotracker Green FM dye to observe all mitochondria despite mitochondrial membrane potential and only TMRE (200 μM) was retro-orbitally injected and incubated for 30 min before imaging. Images were acquired using excitation at 480 nm, 532 nm, and 647 nm.

### Measurements of mitophagy
Mitophagy was measured in eight-to-ten-week-old male mtKeima mice[32] or; C57BL/6J (strain# 000664) injected with Ad-Cox8-EGFP-mCherry ($1 \times 10^9$ PFU diluted in 200 μl of PBS)[33] using intravital microscopy[52]. Lysosome inhibitor leupeptin (80 mg/kg), or saline was injected i.p. 16 h prior to the experiment. A second dose of leupeptin (40 mg/kg) or saline was administered 12 h later. Three to five mice were analyzed per experiment. This leupeptin treatment protocol was used prior to isolating and sorting cells into PP and PC populations for immunoblot analysis of mitophagy proteins.

### Western blot
Proteins from primary hepatocytes or whole liver were extracted by homogenization in RIPA lysis buffer (150 mM NaCl, 0.1% Triton X-100, 0.5% sodium deoxycholate, 0.1% sodium dodecyl sulfate, and 50 mM Tris HCl pH 8.0) containing EDTA, PMSF, and Halt Inhibitor Cocktail (Thermo Fisher), followed by centrifugation at $13,000 \times g$ at 4 °C for 30 min. Protein concentration was determined using a Pierce™ BCA Protein Assay Kit (Thermo Fisher). Lysates were heated at 95 °C for

5 min, and 5–10 µg aliquots fractionated by SDS-PAGE then transferred to nitrocellulose (0.45 µm) membrane (Bio-Rad). Membranes were blocked for 1 h at room temperature in 5% BSA in 1× Tris-buffered saline + 0.1% Tween 20 (TBST), then incubated with primary antibody diluted in 5% BSA in TBST at 4 °C overnight. Membranes were washed three times with TBST then incubated in secondary antibody 1 h at room temperature. Membranes were washed three times with TBST, then Clarity ECL western blot substrate solution (Bio-Rad) was applied for detection and imaged using the ChemiDoc Imaging System (Bio-Rad). Bands were quantified with ImageJ.

### Liver Perfusion and hepatocyte isolation

Livers of anesthetized mice were perfused by inserting a 22-gauge syringe into the portal vein and delivering 25 ml of pre-warmed (37 °C) perfusion buffer [Krebs–Henseleit buffer (Sigma) with 0.5 µM EDTA], followed by 25 ml collagenase A buffer [Krebs–Henseleit buffer, with 0.1 mM $Ca_2Cl$ and 0.4 mg/ml collagenase A (Sigma)]. After perfusion, livers were transferred to a Petri dish, flooded with cold PBS, and hepatocytes were gently released using forceps. Dissociated cells were collected and filtered through a 100 µm cell strainer. Cells were centrifuged at $100 \times g$ for 5 min at 4 °C to obtain hepatocyte-enriched cells. Pellets were resuspended in cold PBS, filtered through a 40 µm cell strainer, and centrifuged at $100 \times g$ for 5 min at 4 °C. Percoll diluted in 10× Hank's buffer (Sigma) was added to the cell suspension and centrifuged at $100 \times g$ for 5 min at 4 °C. The supernatant containing dead cells was removed by aspiration. The pellet was resuspended in cold PBS, and cell number and viability were determined.

### Fluorescence-activated cell sorting (FACS) of PP and PC populations

Cell sorting was performed on a FACSAria Fusion cell sorter (BD Biosciences). Forward and side light scatter was used to distinguish cells from debris and to identify single cells. Zombie-Green fixable live dead dye (1:500, BioLegend) was used to discriminate between live and dead cells. PP and PC populations were identified using anti-E-cadherin-PE and anti-CD73-APC antibody staining (1:150; BioLegend). PP- and PC-specific gates were set after spectral compensation using appropriate fluorescent minus (FMO) controls. The protocol was developed based on ref. 53.

To sort PP and PC hepatocyte populations from mtKeima mice, BD OptiBuild™ BUV395 Rat Anti-Mouse CD324/E-Cadherin (1:125, BD Bioscience) was used to stain PP hepatocyte populations. For E-cadherin negative populations (PC), FSC-A and SSC-A plot was used to further exclude smaller-sized hepatocytes as PC hepatocytes (CD73 positive) was generally found to be bigger in cell size than PP hepatocytes.

### Protein Digestion and TMT labeling

Ad libitum-fed, eight-to-ten-week-old male mice; C57BL/6J (strain# 000664) were used. Cells were lysed in 50 mM HEPES, pH 8.0, 8 M urea, and 10% methanol, followed by sonication. Lysates were clarified by centrifugation, and protein concentration quantified using a BCA protein estimation kit (Thermo Fisher). A 250 ug aliquot was alkylated and digested by incubating overnight at 37 °C in trypsin at a ratio of 1:50 (Promega). Digestion was acidified by adding formic acid (FA) to a final concentration of 1% and desalted using Pierce peptide desalting columns according to the manufacturer's protocol. Peptides were eluted from the column using 50% ACN/0.1% FA, dried in a speed vac, and kept frozen at −20 °C for further analysis.

For TMT labeling, 125 µg of each sample was reconstituted in 50 µl of 50 mM HEPES, pH 8.0, 500 µg of TMTpro-16plex label (Thermo Fisher) in 100% ACN. After incubating the mixture for 1 h at room temperature with occasional mixing, the reaction was terminated by adding 8 µl of 5% hydroxylamine. As there were more than 16 samples, we generated a pooled sample consisting of equal amounts of lysate

from each condition and TMT labeled. The pool was added to each TMT experiment. The TMT-labeled peptides were pooled and dried in a speed vac. The samples were desalted, and excess TMT labels were removed using peptide desalting columns (Thermo Fisher). A 100 µg aliquot of labeled peptide mixture was fractionated using high pH reversed-phase, and the remaining peptide mixture was used for phospho-enrichment.

### Phosphopeptide enrichment

Phosphopeptides were enriched from the TMT-labeled peptides using the SMOAC (Sequential enrichment of Metal Oxide Affinity Chromatography) method. First, the phosphopeptides were enriched using the HiSelect $TiO_2$ phosphopeptide enrichment kit (Thermo Fisher) according to the manufacturer's protocol. The flow-through and wash from the $TiO_2$ enrichment kit were then combined and enriched using the HiSelect Fe-NTA phosphopeptide enrichment kit (Thermo Fisher). The enriched phosphopeptides obtained by both methods were dried by speed vac and stored at −20 °C until analysis by mass spectrometry.

### High pH reversed-phase fractionation

The first-dimensional separation of peptides was performed using a Waters Acquity UPLC system coupled with a fluorescence detector (Waters) using a 150 mm × 3.0 mm Xbridge Peptide BEM™ 2.5 µm C18 column (Waters) operating at 0.35 ml/min. The dried peptides were reconstituted in 100 µl of mobile phase A solvent (3 mM ammonium bicarbonate, pH 8.0). Mobile phase B was 100% acetonitrile (Thermo Fisher). The column was washed with mobile phase A for 10 min followed by gradient elution 0–50% B (10–60 min) and 50–75% B (60–70 min). Fractions were collected every minute. These 60 fractions were pooled into 24 fractions. Fractions were vacuum centrifuged, and lyophilized fractions were stored at −80 °C until analysis by mass spectrometry.

### Mass spectrometry acquisition

The lyophilized peptide fractions were reconstituted in 0.1% TFA and subjected to nanoflow liquid chromatography (Thermo Ultimate™ 3000RSLC nano LC system, Thermo Fisher) coupled to an Orbitrap Eclipse mass spectrometer (Thermo Fisher). Peptides were separated using a low pH gradient and 5–50% ACN over 120 min in a mobile phase containing 0.1% formic acid at a 300 nl/min flow rate. MS scans were performed in the Orbitrap analyzer at a resolution of 120,000 with an ion accumulation target set at $4e^5$ and max IT set at 50 ms over a mass range of 400–1600 m/z. Ions with a determined charge state between 2 and 5 were selected for MS2 scans in the ion trap with CID fragmentation (Turbo; NCE 35%; maximum injection time 35 ms; AGC $1 \times 10^4$). The spectra were searched using the Real-Time Search Node in the tune file using mouse Uniprot database using Comet search algorithm with TMT16 plex (304.2071 Da) set as a static modification of lysine and the N-termini of the peptide. Carbamidomethylation of cysteine residues (+57.0214 Da) was set as a static modification, while oxidation of methionine residues (+15.9949 Da) was set up as a dynamic modification. For the selected peptide, an SPS–MS3 scan was performed using up to 10 b- and y-type fragment ions as precursors in an Orbitrap at 50,000 resolution with a normalized AGC set at 500 followed by maximum injection time set as "Auto" with a normalized collision energy setting of 65.

The enriched phospho-samples were reconstituted in 0.1% TFA and analyzed using an Orbitrap Eclipse mass spectrometer. Peptides were separated using a low pH gradient, and FAIMS was enabled during data acquisition with the compensation voltages set at −45V, −60V, and −75V set up at the first run, −50V, −65V, and −80V for the second run and −55V, −70V and −85V for the last run. MS scans were performed in the Orbitrap analyzer at a resolution of 120,000 with an ion accumulation target set at $4e^5$ and max IT set at 50 ms over a mass range of 350–1600 m/z. Ions with a known charge state between 2 and

6 were selected for MS2 scans. A cycle time of 1 s was used for each CV, and a quadrupole isolation window of 0.4 m/z was used for MS/MS analysis. An Orbitrap at 15,000 resolution with a normalized AGC set at 250 followed by maximum injection time set as "Auto" with a normalized collision energy setting of 38 was used for MS/MS analysis. The node "Turbo TMT" was switched on for the high-resolution acquisition of TMT reporter ions.

## Data analysis

Acquired MS/MS spectra were searched against the mouse Uniprot protein database and a contaminant protein database using SEQUEST in the Proteome Discoverer 2.4 software (Thermo Fisher). The precursor ion tolerance was set at 10 ppm, and the fragment ions tolerance was set at 0.02 Da, with methionine oxidation included as a dynamic modification. For the phospho-enriched samples, serine, threonine, and tyrosine phosphorylation were set as variable modifications. Carbamidomethylation of cysteine residues and TMTpro16 plex (304.2071 Da) was set as static modification of lysine and the N-terminus of the peptide. Trypsin was specified as the proteolytic enzyme, with up to 2 missed cleavage sites allowed. Searches used a reverse sequence decoy strategy to control for the false peptide discovery, and identifications were validated using the Percolator algorithm in Proteome Discoverer 2.4.

Reporter ion intensities were adjusted to correct for lot-specific impurities according to the manufacturer specification, and the abundances of the proteins were quantified using the summation of the reporter ions for all identified peptides. A two-step normalization procedure was applied to a 24-plex TMT experiment [2 TMT experiments with 12 samples each]. First, the reporter abundances were normalized across all the channels to account for equal peptide loading. Secondly, the intensity from the pooled sample was used to normalize the batch effect from the multiple TMT experiments. Samples were further normalized using the Voom algorithms and quantile normalization from the Limma R package (v3.40.6). We evaluated the significance of a given protein, phosphopeptide, or normalized phosphopeptide change using Limma, where zonation was determined by the $p$-value (0.05) and/or FC ($\geq$1.2). Pathway enrichment analysis was performed using Fisher's Exact test with GO database. MitoCarta 3.0 was used to classify of the identified proteins and phosphoproteins into mitochondrial and non-mitochondrial proteins. GraphPad Prism 9 and BioRender were used for visualization.

## Prediction of mTOR and AMPK activity using a Group-based Prediction System (GPS)

All peptides with PP or PC zonation were extrapolated as 15 AA sequences with the phosphosite in the center. Peptides without a specific site were excluded. Duplicate sequences were removed, and sequences were analyzed using GPS 5.0 (AMPK and mTOR selected) at a medium threshold. The sequences analyzed and the output from GPS are listed in Supplementary File 6.

## MoMo motif analysis

Motif analyses of PP or PC phosphopeptides were performed using the web tool available at: https://meme-suite.org/meme/tools/momo, v5.5.4. All identified 15 AA sequences without duplicates for each PP and PC quantified phosphosite were analyzed with a $p$-value threshold at $10^{-6}$. The results from this analysis are listed in Supplementary File 4.

## WNT-activated genes correlation

We examined identified mitochondrial proteins associated with WNT signaling transduction[54]. PP and PC were separately analyzed by computing the $\log_2$ ratio of the mean protein abundance over the geomean from the raw data. These values were then used to examine Spearman correlation value with log2 ratio of the mean gene expression levels from WNT-hyperactivating liver-specific APC knockout and control mouse liver samples.

## Seahorse assay

A Seahorse XFe96 Analyzer and XF Mito Stress Test Kit (Agilent Technologies) were used to measure the oxygen consumption rate (OCR) and extracellular acidification rate (ECAR) of PP and PC-sorted cells from ad libitum-fed, eight-to-ten-week-old male mice; C57BL/6J (strain# 000664). A day before the assay, the cartridge sensor was hydrated overnight with Seahorse Bioscience Calibration Solution at 37 °C without $CO_2$. After sorting, cells were seeded at $5.0 \times 10^4$ cells/well, in a 96-well seahorse plate and allowed to adhere for 2 h. After 2 h, media was replaced with serum-free Seahorse XF Base media containing glucose, pyruvate, and glutamine (pH 7.4), and cells were incubated at 37 °C in a non-$CO_2$ incubator for 1 h. OCR and ECAR were measured after the injection of oligomycin (1.5 μM), FCCP (1 μM), and rotenone/antimycin (0.5 μM). The XF Mito Stress Test Kit was used together with an XF MitoFuel kit to test fuel dependency. Three inhibitors, UK5099 (2 μM), BPTES (3 μM), and etomoxir (4 μM), were added to measure the dependency of cells on the three primary mitochondrial fuels to produce ATP; glucose, glutamine, and long-chain fatty acids, respectively. Average protein amount of PP and PC seeded wells were examined with BCA for normalization. To overcome reading variations within each population, and measure the effect of the inhibitors, the measurements of the wells treated with inhibitors were further normalized to the first basal reading of the group treated with the vehicle (control). Measurements for ATP production rate and maximum respiration capacity were presented as a percent normalized to PP. Bar graph shows data from four independent experiments presented as mean ± SD. Each independent experiment was conducted with five to twelve replicates per population and treatment.

## ATP measurement assay

The bioluminescence ATP assay kit was used according to manufacturer instructions (Invitrogen) to analyze spatially sorted hepatocytes. ATP levels were calculated and normalized according to protein amount.

## Citrate Synthase activity assay

Citrate Synthase (CS) activity was analyzed following the manufacturer protocol (Abcam). After collecting the cells via FACS, cells were lysed. A 50 μl aliquot of supernatant or standard GSH solution with 50 μl of reaction buffer was added per well in a 96-well plate to measure the absorbance (OD 412 nm) in kinetic mode at 25 °C for 40 min. CS activity was calculated based on two different time points considering the dilution factor and cell number, then normalized to the PP fraction.

## Triglyceride (TG) measurement assay

After cell isolation and FACS, the sorted PP and PC cells were pelleted, washed twice with cold PBS, and lysed on ice for 30 min. Lysates were centrifuged at 13,000 × $g$ for 10 min at 4 °C and the supernatant collected. A TG-colorimetric GPO-PAP assay kit (Elab Science) was used to determine TG content, following manufacturer instructions. TG levels were calculated according to protein amount and normalized to PP fraction.

## DNA and RNA isolation

RNA was isolated from spatially sorted cells using the RNeasy mini kit and QIAshredder homogenizer columns (Qiagen). Total DNA was isolated using the DNeasy Blood and Tissue kit (Qiagen).

## Quantitative PCR

A High-Capacity RNA to cDNA kit (Thermo Fisher) was used to synthesize random-primed cDNA from 600 ng DNAse treated RNA. Real-

time PCR was conducted in 384 well-plates using a ViiA7 Real-Time PCR system (Applied Biosystems). Singleplex reactions (10 µl) containing a FAM-MGB expression assay for the gene of interest (Acaca Mm01304289_m1, Acly Mm01302282_m1, Fasn Mm00662319_m1, Scd1 Mm00772290_m1) or endogenous control (Ppia Mm02342430_g1) (Thermo Fisher) were performed using cDNA synthesized from 3 ng RNA and 1× Fast Advanced Master Mix (Thermo-without Amp Erase UNG). The comparative Ct method (delta, delta Ct) was used to determine relative expression normalized to Ppia (Applied Biosystems® ViiA™ 7Real-Time PCR System Getting Started Guides).

## mtDNA copy number determination

Mitochondrial DNA content was measured by comparing the DNA copy number of two mitochondrial-encoded genes (ND1 and 16S rRNA) to HK, a nuclear-encoded gene[55]. The mitochondrial to nuclear DNA ratio (mtDNA/nDNA) was determined by Real-Time PCR using an ABI Viia7 Real-Time PCR System (Applied Biosystems). Because mitochondrial genes lack introns, it was possible to use TaqMan gene expression assays for mtDNA copy number. Singleplex reactions (10 µl) containing FAM-MGB assays for ND1 Mm04225274_s1, Rnr2 Mm04260181_s1, or HK2 Mm00193901_cn (Applied Biosystems) were performed in quadruplicate using 384 well-plates with 10 ng DNA and 1× Universal Master Mix (Applied Biosystems-without Amp Erase UNG). Determination of mtDNA content was performed using the comparative Ct method (delta, delta Ct) with normalization to HK2 as an internal reference (Applied Biosystems® ViiA™ 7Real-Time PCR System Getting Started Guides).

## Drug treatment and flow cytometry assay

Mice were treated with two doses of either vehicle, Compound C (CpC, 20 mg/kg), 5-Aminoimidazole-4-carboxamide ribonucleotide (AICAR, 0.5 mg/kg), Torin I (20 mg/kg) or MHY1485 (20 mg/kg) before primary hepatocytes were isolated via perfusion. Hepatocytes were then labeled with Cy-PE/PE-anti-E-cadherin and APC-anti-CD73 and either JC1 (5 µM) or BODIPY-493/503 (2 µM) for 20 min at 37 °C. The fluorescent signal was measured by BD LSRFortessa (BD Biosciences) cell analyzer. Consistent with FACS, forward and side light scatter was used to distinguish cells from debris and to identify single cells.

## Sample prep and imaging for FIB-SEM

The structured sample preparation protocol based on the sample preparation widget from EMPIAR (https://www.ebi.ac.uk/empiar/widget) can be accessed at: 10.5281/zenodo.8422314. Briefly, lower left lobes dissected from ad libitum-fed, eight-to-ten-week-old male mice; C57BL/6J (strain# 000664) were fixed in Karnovsky's fixative for 2 h at room temperature, rinsed (all rinses were typically five times 3 min each at room temperature) in 0.1 M sodium cacodylate buffer, then fixed in 2% w/v osmium tetroxide + 1.5% w/v potassium ferricyanide in sodium cacodylate for 1 h at room temperature. The biosamples were then rinsed in dd-water, then en bloc stained with 1% uranyl acetate in dd-water overnight at 4 °C. These were then rinsed in dd-water and incubated in freshly made lead aspartate for 30 min at 60 °C. After another dd-water rinse, the biosamples were dehydrated for 10 min each in 35%, 50%, 70%, 95%, three times in 100% ethanol, and finally in three times in 100% propylene oxide. The dehydrated tissues were infiltrated with Polybed 812 resin (hard formulation) in resin: PO ratios of 1:3 (1 h), 1:1 (overnight), 3:1 (5 h), and finally 100% resin overnight. These were transferred to beam capsules and the resin was cured at 60 °C for 48 h to yield samples ready for vEM. Liver samples were sectioned and stained with Toluidine Blue to identify periportal and pericentral regions in the tissue. The remaining sample was trimmed and mounted on an SEM stub, and after a thin coat (approximately 8 nm) of carbon was sputtered, the specimens were into a FIB-SEM (Zeiss crossbeam 550). The same areas were located using SEM imaging, and Fields of View (FOVs) for vEM imaging were defined.

Once the target cell was identified, the Fibics Atlas 3D FIB sample preparation workflow was initiated. A 1um thick protective platinum pad was deposited over the sample using an FIB current of 1.5 nA, after which tracking and autofocus lines were milled into the platinum surface and subsequently covered by an FIB-mediated deposition of 1 um carbon at 1.5 nA. A coarse trench was milled using a 30 nA FIB beam and fine polished using a 3 nA beam. Milling and imaging parameters during the "continuous mill-and-image" acquisition run were set at 30 kV accelerating voltage, 1.5 nA FIB current, and 1.5 kV accelerating voltage, 1.1 nA SEM beam current. A pixel resolution of 5 nm with a 10 nm slice thickness was set, with total dwell time of 3.0 us/pixel, and the EsB detector grid voltage was set at 825 V. Image frame time was approximately 42 s, FIB advance rate was approximately 13.2 nm/min. The raw image stacks were then registered, inverted, and binned using in-house scripts[56].

## Mitochondria segmentation and image analysis

Volume EM 3D reconstructions were imported into napari (https://napari.org), and mitochondria were segmented using the empanada plugin (https://empanada.readthedocs.io/en/latest/empanada-napari.html) by deploying the deep learning model MitoNet-mini with standard presets and ortho-inference[26]. The model produced high-quality mitochondrial segmentations, with some split and merge errors as expected for crowded mitochondria. These errors were manually corrected using empanada to produce instance segmentations of mitochondria from PP and PC volumes. Mitochondria that were partially cut off by the image limits were computationally removed to prevent artifactual measurements. The surface area, volume, and sphericity of remaining mitochondria (PP: $n = 175$, PC: $n = 250$) were measured for each instance using in-house scripts based on the scikit-image library (https://scikit-image.org/).

## Statistics & reproducibility

To verify the reproducibility of results, we ran a minimum of three independent experiments, which resulted in similar results. Every sample quantified relates to an individual mouse, so all samples are biological replicates. Additional information on the number of samples or mice used in each experiment is included in the figure legends.

## Reporting summary

Further information on research design is available in the Nature Portfolio Reporting Summary linked to this article.

# Data availability

The mass spectrometry dataset has been deposited in the MassIVE database under accession code MSV000093282 [https://massive.ucsd.edu/ProteoSAFe/static/massive.jsp]. The remaining data are available within the Article, in the Supplementary Information, Supplementary Data 1-4, Supplementary Movies 1-4, and in the Source Data file. Source data are provided with this paper.

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

## Acknowledgements

We thank Dr. Wen-Xing Ding for the Cox8-EGFP-mCherry-AAV8 construct. Drs. Torn Finkel and Nuo Sun for mtKeima mice. We thank Drs. Sudipto Das and Thorkell Andresson from the CCR proteomics Facility at the Frederick National Laboratory for Cancer Research and members of the NCI LGI Flow Cytometry Core. We thank Aayush Bhatawadekar and Abhishek Bhardwaj for volume renderings and morphological calculations based on vEM mitochondrial segmentation. Dr. Daniel Feliciano acquired images in Fig. 4A and Supplementary Fig. 2A. We thank Pranali Pathare Mangat of 3P Scientific Communications for providing scientific editing support. We are grateful for the critical comments made by members of the Porat-Shliom lab. This work was supported by the Intramural Research Program at the NIH, National Cancer Institute 1ZIABC011828 (NPS), and under Contract No. 75N91019D00024 (KN). The content of this publication does not necessarily reflect the views or policies of the Department of Health and Human Services, nor does mention of trade names, commercial products, or organizations imply endorsement by the U.S. Government. The authors have no conflicts to report.

## Author contributions

SWSK and NPS designed the research; SWSK, RPC, CBM, LAB, and CMC performed the research; SWSK, RPC, CMC, and NPS analyzed the data. AH and KN performed the FIB-SEM and associated quantification. LMJ, AL, and MC provided bioinformatics expertise for data analysis. JH provided the human samples. NPS wrote the manuscript, with input from all co-authors.

## Competing interests

The authors declare no competing interests.
