## [Peer Review File · Nature Communications]

REVIEWER COMMENTS

Reviewer #1 (Remarks to the Author):

Kang et al explore the differences in mitochondrial mass and shape in different zones of the liver, specifically in the periportal (PP) and pericentral (PC) regions of liver. Several novel findings are reported including identification of mitochondrial proteins that are differentially expressed and differentially phosphorylated in PC versus PP zones of the liver. The authors also show that, compared to PC mitochondria, PP mitochondria are smaller and more numerous and have higher membrane potential, consume more oxygen, produce more ATP and oxidize fatty acids. By contrast, PC mitochondria are more tubulated and engage in lipid synthesis. The authors suggest somewhat counterintuitively (given there are more smaller mitochondria and lower expression of mitophagy regulators) that PP hepatocytes have higher rates of mitophagy than PC hepatocytes and that differences in mitochondrial volume are regulated by AMPK and mTOR signaling. While the proteomics and phosphoproteomics provide interesting data and the imaging of mitochondria at high resolution and in vivo is highly commended, there are concerns that need to be addressed, as detailed below.

Major points

1. The Mitotracker staining in figure 3A is not uniform or homogeneous – nor would one expect it to be if there are the reported differences in mitochondrial mass and mitophagy described later in the manuscript so it should be clarified on line 91 that there are differences.
2. In the same figure, it looks more like the TMRE staining is in zone 2 and not PP. It would greatly assist analysis if this figure was shown at higher magnification similar to liver images in figure 1B and also that the defining features of PP and PC were more clearly visible.
3. The authors should note that lipid levels are not simply due to lipogenesis, but could be explained by altered oxidation rates, altered uptake of lipid, decreased lipophagy etc. So the authors should be more circumspect about comments made in line 112. Also, for Fig. S2, it should be noted that the lipogenesis genes noted (Fasn, Acly, Acaca, Scd1) are subject to negative feedback control via lipid-controlled regulation of Srebps so one would expect to see them up-regulated at the transcriptional level in regions where there is less lipid so the comment in lines 117 – 119 needs to be amended.
4. The area of most concern pertains to the analysis of mitophagy. Up to figure 5, all data points to there being more mitochondria, smaller mitochondria and increased OCR/ATP generation in PP regions which in turn would suggest decreased mitophagy in the PP region since, as the authors mention, mitophagy preferentially eliminates smaller mitochondria with fused mitochondria spared. Plus, as the authors show, protein expression of Bnip3, Bcl2l13, Gabarapl3 and Gabarap are all increased in PC regions (Fig. 5C). Plus, if you look at Fig. 5B, you see increased mitophagy with Leupeptin treatment most in the PC region while there is minimal mitophagic flux in the PP region, at least based on the histogram in Fig. 5B. It would be good to see the actual images for this data to compare to Figure 5A. It seems as if there may just not be very much mitophagy at all in the liver shown in Figure 5A. Indeed, all of this is done under

fed conditions, and liver mitophagy is most induced by fasting of mice. The authors should show mt-mKeima images for livers from fed and fasted mice and plus/minus leupeptin and graph accordingly.

5. Turning to SFig.4 which also addresses mitophagy, the text on line 167 claims that Sfig.4A shows “accumulation of mitochondria in lysosomes” which it really does not – it is a western blot of processed LC3 from whole liver from fed mice plus/minus leupeptin. Images of mitochondria inside lysosomes would indeed be very useful here but that is not what is shown so the text needs correcting or images of “mitochondria in lysosomes” added to the panel. Also if showing LC3 processing, would be best to perform on lysates from PP versus PC, not from whole liver.

6. The images in SFig.4B and graphed in SFig.4C have been mis-interpreted. Again, there is more flux in the PC region since there is more of an increase in mitophagosomes when leupeptin is added, as is clear from the graph Fig.S4C) – leupeptin really does not change the levels of mitophagosomes in the PP region but does increase them in the PC region which argues that there is more mitophagic flux in the PC region and this is also more consistent with the authors other observations that there are more mitochondria, smaller mitochondria and increased OCR and ATP generation in the PP region. And while the PC hepatocyte starts out with fewer mitochondria (Fig.S4C bottom left), that is as likely due to increased mitophagy in vivo that has already taken place since mitophagosomes accumulate in the PC hepatocyte when they block mitophagy with Leupeptin (Fig.S4C bottom right) indicating very nicely that there is more mitophagy in PC cells (blue bars in graph in Fig.S4C increase with Leupeptin illustrate this) than in PP cells where Leupeptin has minimal effect (red bars in graph in Fig.S4C do not change). Overall, the assessment of there being increased mitophagy in PP regions and less mitophagy in PC regions is incorrect and needs to be re-assessed since the data presented do not support this conclusion. To the contrary, the data support the converse. To make a more robust claim about rates of mitophagy, the authors should also perform double labeling for mitochondria and lysosomes (like they do in Figure 1 for CD73 and E-cadherin), perhaps CoxIV and Lamp1 co-staining, plus or minus Leupeptin. Importantly, given that mitophagy and autophagy more generally is induced by fasting, the authors should add data examining the effect of fasting on their observations.

7. The second area of concern is the interpretation that increased phosphorylation of proteins like Mtor1, Mtor2, Bnip3 and Bcl2l13 results in their activation, as mentioned lines 194 to 197 and 295 - 297. The phosphorylation of these proteins could be inhibitory and this is not at all considered. The interpretation and text here needs to be reconsidered.

8. The authors make claims about AMPK and mTOR activation without measuring zonal differences in their activity. The westerns in Figure S6 need to be repeated on sorted PP and PC hepatocytes. In addition, they should examine how zonal activity of AMPK and mTOR changes in response to fasting.

9. Finally, the graph in figure 7B need to be revisited since if you look at the effects of AMPK and mTOR activators/inhibitors, you can see here again that the blue (PC) numbers are the ones that change the most, and again are more consistent with there being more mitophagy in the PC regions not the PP regions, not less.

Minor points

1. Please define GS in figure 1D in the figure legend.

2. The text in Figure 2C is too small to be legible.

Reviewer #2 (Remarks to the Author):

This is a very interesting study on mitochondrial heterogeneity within liver tissue. The paper focuses on characterizing mitochondria from two spatially distinct populations of hepatocytes – those in the periportal (PP) region as well as those in the pericentral (PC) region. Previous literature has documented liver zonation, but studies have focused either on gene expression differences or morphological distinctions (such as those by electron microscopy). While suggestive, these data do not fully reflect the heterogeneity seen in liver tissue, particularly at the level of mitochondrial function. This study addresses this gap by performing thorough and impressive characterization of hepatocyte mitochondrial function in isolated organelles, primary cells, and in vivo.

The study begins with a proteomics analysis on spatially sorted hepatocytes, with PC cells expressing the marker CD73 and PP cells expressing the marker E-cadherin. The immunofluorescent imaging of these regions in Figure 1B is particularly striking. The results from the proteomics experiment suggest that hundreds of mitochondrial proteins are ‘zonated,’ or preferentially expressed in either PP or PC hepatocytes relative to the other cell type. This is extremely interesting. Many studies have suggested mitochondrial heterogeneity across tissues, but this, to my knowledge, is amongst the first demonstrations of such striking proteomic diversity across the same cell type in a singly tissue. Based on these data, the authors hypothesize that PP mitochondria will have greater oxidative capacity relative to PC mitochondria, and, using a variety of techniques both in vivo and in isolated cells, show significant differences in the metabolic programs of PP and PC hepatocytes. Specifically, the authors propose that PP mitochondria are better suited for beta oxidation, while PC mitochondria are more specialized for lipid synthesis. The authors continue in their characterization of PP and PC mitochondria, finding that PP mitochondria are morphologically distinct (i.e., they have distinct sizes and shapes) relative to PC mitochondria, and that PP mitochondria display higher rates of mitophagy. Finally, the authors perform phosphoproteomic analysis of PP and PC hepatocytes. The authors identify numerous phosphorylation events on mitochondrial proteins across various pathways, including mitophagy and fusion/fission, that are spatially enriched in one of the two cell types. The authors then end the manuscript by determining that two key nutrient sensing kinases, AMPK and mTOR, can modulate a subset of the phenotypes described in PP and PC hepatocytes. Overall, the authors use a wide breadth of experimental techniques to characterize mitochondrial heterogeneity in mammalian liver tissue. In my opinion, this study is highly innovative and provides not only key techniques to the field (e.g., spatial sorting of hepatocytes), but will provide key datasets for many follow up studies.

Despite these strengths, as well as the rigor in many of the experiments performed in this manuscript, there are some aspects of this study that would be strengthened through additional experimentation to support key claims (see below). I include my concerns and potential experimental suggestions to address these concerns below (see 'Major revisions'). I also include some minor points that would make the manuscript easier to read and/or understand (see 'Minor revisions').

Major revisions

Major revision #1: Addressing claims

This paper puts forth a number of claims (text sourced from the paper abstract), most of which are rigorously tested, including:

1. "...PP and PC mitochondria are morphologically and functionally distinct..."
2. "...beta-oxidation and mitophagy were elevated in PP regions..."
3. "...acute pharmacological modulation of nutrient sensing through AMPK and mTOR shifted mitochondrial phenotypes in the PP and PC regions of intact liver."

However, two claims that may be overstated or should be reworded are as follows:

4. "...lipid synthesis was predominant in the PC mitochondria."

- The authors put forth data consistent with lipid synthesis being elevated in PC mitochondria relative to PP mitochondria, but do not directly test this. (See Major Revision 2 for suggestions on addressing this.)

5. "...mitophagy and lipid synthesis are regulated by phosphorylation in a zoned manner."

- The authors indeed find that proteins involved in mitophagy and lipid synthesis have distinct, zoned phosphorylation patterns (Figures 6G and H), but they do not put forth any evidence to suggest that these events are regulatory. Such a claim would require more data to be put forth regarding the specific

function and the dynamic contexts in which these phosphorylation events are altered. The generation of such data is likely beyond the scope of this manuscript, and I suggest that the authors reword this claim rather than perform extensive follow up experimentation on this point.

-

Major revision #2: More rigorous testing of lipid synthesis in zoned hepatocytes

As noted above, the authors claim that lipid synthesis is predominant in PC mitochondria. The data supporting this claim are:

- a. significant elevation of lipid droplet (LD) formation in PC hepatocytes (Figure S2A),
- b. elevated citrate synthase (Cs) activity in PC hepatocytes (Figure 3J), and
- c. elevated TG levels in PC hepatocytes (Figure 3K).

The authors put forth this collection of data to suggest that PC hepatocytes are more lipogenic, and the data are consistent with such a model. However, the data are also consistent with other models. For instance, elevated LDs, as well as elevated TGs (which are likely coupled phenotypes) could be derived from an increase in lipid synthesis (as proposed), decreased LD catabolism (including decreased FAO, which they authors indeed show in Figure 3F), or impaired TG/VLDL secretion (reviewed in PMID: 28428634). Thus the increase in LDs/TGs is not sufficient to claim bona fide lipogenesis.

Second, the authors claim that Cs activity is increased in PC hepatocytes, however no data on Cs total protein expression is presented. Is it possible that Cs is simply overexpressed in PC hepatocytes relative to PP hepatocytes? It is also well known that Cs activity correlates to total mitochondrial content, particularly in liver as recently tested (PMID: 33077793). While these experiments may still prove consistent with the model that Cs activity is elevated due to lipogenesis, the point in bringing up these other studies is that Cs activity could be altered independent of lipogenesis, and thus is an indirect measure at best.

As one of the focal points of the paper is the nutrient regulation of PP versus PC hepatocytes, the authors should perform functional lipogenesis experiments in both PP and PC hepatocytes to demonstrate that lipid synthesis is elevated in PC relative to PP cells. Experimentally, there are many protocols to assay lipogenesis in primary hepatocytes, including PMID: 26382148. The authors could also bolster this claim by assaying LD formation in a fasting/refed condition – one would expect to see increases in LD formation in this assay in PC versus PP hepatocytes in post-prandial conditions (although admittedly this is an indirect assay). The authors could also mine their phosphoproteomics analysis for

well described lipogenic PTMs such as Acc phosphorylation (gene symbol Acaca; phospho-S79 – inhibitory and would be expected to be down in PC hepatocytes if lipogenesis is up; this one is also of particular interest because it is regulated by Ampk) or Acly (phospho-S455 – activating and would be expected to be up in PC hepatocytes).

Major revision #3: Further exploring mitophagy phenotypes and proteins

The authors show very interesting data that mitophagy flux increases in PP mitochondria despite the fact that mitophagy proteins such as Bnip3 and Bcl2l13 are enriched in PC mitochondria. The authors propose this is because “basal mitochondrial is higher in the metabolically active PP hepatocytes, which may maintain the levels of the mitophagy-related proteins low.” This could be tested with the leupeptin experiment shown in Figures 5A/B; if mitophagy proteins are kept low because of high flux, they should accumulate in the presence of a lysosomal blocker. The authors should test this, with any means that is easiest for them to detect mitophagy proteins. (Bnip3 or Bcl2l13 would be preferable to test here because they are reported in the dataset, but others would suffice for this line of experimentation as well.)

Major revision #4: Maximizing impact with proteomic and phosphoproteomic datasets

a. The authors perform TMT-based proteomics and phosphoproteomics as a part of the characterization of PP and PC hepatocytes in this manuscript. Unfortunately, the raw datasets were not provided to the reviewers, so I could not look into the trends with the data. Upon publication, these datasets should be provided as supplementary information so as to provide an accessible resource for readers of this manuscript. Additionally, the raw datasets should be deposited into a database such as PRIDE.

b. The proteomics and phosphoproteomics data in PP versus PC hepatocytes are visualized appropriately and show interesting trends. However, I believe the relative protein abundance changes could be confirmed for a handful of targets by a second measure. This not because I do not believe the proteomics data, but rather because proteins quantified by proteomics, particularly with TMT tags, are prone to dynamic range compression due to precursor interference (see PMID: 24211767). It is thus possible that the differences in protein expression between the two cell types are being underestimated. Immunofluorescence similar to what is shown in Figure 1B for a couple of differentially expressed targets would be powerful validation.

c. The schematics in Figure 2C and D would be improved with some type of metric for the fold change between the two cell types (if possible. This is a minor comment.)

d. Some of the phosphosites in Figure 6H seem to be mislabelled. For instance, Bnip3 is annotated as being phosphorylated at “S78” but residue 78 is a Hisidine (Uniprot: O55003); similarly, Mtfr1 is phosphorylated at “S234/235” but these residues are Pro, Gln (Uniprot: Q99MB2) and Mtfr1l is phosphorylated at “T118” but this is an alanine (Uniprot: Q9CWE0). Bcl2l13, Pkraca, and Apex1 have correct residues at their annotations, but that is as far as I validated. The authors should recheck this; this could also be alleviated by including the identified peptide within a supplemental table (as pointed out in 3a).

e. The authors perform functional experiments with AMPK and mTOR activators/inhibitors and find meaningful changes between PP and PC hepatocytes, but they do not tie this back into the phosphoproteomics analysis. Are there phosphosites that lie within AMPK or mTOR consensus motifs? If so, does this suggest direct regulation of these substrates, or do the authors propose that this is indirect regulation? Such points should be brought up in the discussion (beyond the single example of AMPK-mediated regulation of Mtfr1l. Also, does this citation characterize the same Mtfr1l phosphosite identified in this paper? If so, this should be highlighted.).

f. Pdha1 phosphorylation is elevated in PC hepatocytes (Figure 6H), consistent with decreased flux through pyruvate dehydrogenase. This is somewhat at odds with data presented in Figure 3H showing that PC hepatocytes are more susceptible to loss of ATP in the presence of UK5099, an inhibitor of pyruvate uptake. The authors should comment on this.

Minor revisions:

1. Cellular ATP is presented as a normalized percentage in Figure 3I; this data would be stronger presented as absolute values.

2. The authors note that PP hepatocytes are smaller than PC hepatocytes, and that modulation of mTOR can change some of these effects. mTOR is well known to control cell size (PMID: 12080086) – do the authors see key targets of mTOR such as S6 differentially phosphorylated in PP versus PC hepatocytes that could contribute to such phenotypes?

Manuscript NCOMMS-23-17889

A spatial map of hepatic mitochondria uncovers functional heterogeneity shaped by nutrient-sensing signaling

We thank the reviewers for their insightful comments, which we feel have **substantially** improved the manuscript and the interpretation of the findings.

Several major revisions were made in the resubmitted manuscript:

1. Following submission, we performed FIB-SEM to gain higher-resolution images of mitochondria for morphological analysis. While parameters like sphericity and overall morphology (spherical versus tubular) were also observed with FIB-SEM, we found a discrepancy between mitochondrial volume measurements obtained with this method compared with light microscopy (Figure 4G and 4I in the original submission). We believe this is due to the quenching of the MitoTracker green. Quenching occurs at high concentrations of charged dyes (e.g. MitoTracker Green) leading to the reduction of the fluorescent signal, rather than an increase. We believe that the high membrane potential in PP mitochondria led to the accumulation of MitoTracker Green and quenching of the signal, giving a false impression that PP mitochondria had a lower volume. FIB-SEM measurements, on the other hand, demonstrated that PP mitochondria have larger volume and surface area than PC (New Figure 4D and E). Therefore, any quantitative analysis using MitoTracker Green fluorescence intensity was removed (Figures 4G, 4I, and 7B in the original submission) and replaced with FIB-SEM analysis (revised Figure 4B-F and new Figure S5). In Figures 3A MitoTracker Green was used for qualitatively labeling mitochondria only.
2. We added substantial information to clarify the rationale behind the experiments modulating mTOR and AMPK signaling. In brief, liver zonation was shown to be driven by differential gene expression, primarily through Wnt/ β -catenin. Our goal was to 1. Characterize mitochondria functionally and 2. Investigate whether mitochondrial diversity was impacted by zoned expression (developmentally or regulated by Wnt/ β -

catenin) or metabolically (dynamically adapting to changes in nutrients). The finding that the mitochondrial phosphoproteome was zoned, and that many of the phospho-sites were responsive to refeeding, insulin, mTOR, and AMPK led to the hypothesis that gradients of nutrients along the PP-PC axis shape mitochondrial diversity through phosphorylation. To uncouple transcription from phosphorylation, we chose a pharmacological approach to acutely target kinases involved in nutrient sensing, rather than fasting the mice which is accompanied by a transcriptional response.

3. New authors were added to acknowledge their contribution to the revision work.

Below is a point-by-point response to individual comments made by the reviewers (blue fonts). Major revisions in the main text were highlighted in yellow.

Reviewer #1

Kang et al explore the differences in mitochondrial mass and shape in different zones of the liver, specifically in the periportal (PP) and pericentral (PC) regions of the liver. Several novel findings are reported including identification of mitochondrial proteins that are differentially expressed and differentially phosphorylated in PC versus PP zones of the liver. The authors also show that, compared to PC mitochondria, PP mitochondria are smaller and more numerous and have higher membrane potential, consume more oxygen, produce more ATP and oxidize fatty acids. By contrast, PC mitochondria are more tubulated and engage in lipid synthesis. The authors suggest somewhat counterintuitively (given there are more smaller mitochondria and lower expression of mitophagy regulators) that PP hepatocytes have higher rates of mitophagy than PC hepatocytes and that differences in mitochondrial volume are regulated by AMPK and mTOR signaling. While the proteomics and phosphoproteomics provide interesting data and the imaging of mitochondria at high resolution and in vivo is highly commended, there are concerns that need to be addressed, as detailed below.

We are thankful for the reviewer's appreciation of the study.

Major points

1. The Mitotracker staining in figure 3A is not uniform or homogeneous – nor would one expect it to be if there are the reported differences in mitochondrial mass and mitophagy described later in the manuscript so it should be clarified on line 91 that there are differences.

The statement was removed.

2. In the same figure, it looks more like the TMRE staining is in zone 2 and not PP. It would greatly assist analysis if this figure was shown at higher magnification similar to liver images in figure 1B and also that the defining features of PP and PC were more clearly visible.

This is an important point. To examine the spatial positioning of mitochondria with high membrane potential, we performed an Intravital microscopy (IVM) experiment labeling mitochondrial membrane potential with TMRE and PP hepatocytes using fluorescently conjugated E-cadherin antibody, as previously described by our group (PMID: 34033261). We now show in a new Figure S3 that hepatocytes displaying high membrane potential are also positive for E-cadherin, indicating they are located in PP regions (zone 1). Per the reviewer's request for an image comparable to Figure 1B, we would like to highlight the difference between imaging of fixed liver sections (Figure 1B) and intravital microscopy (IVM) of the liver in live, anesthetized mice (Figures 3A and the new Figure S3). When imaging intact tissue in a breathing mouse, there is always some motion due to blood flow and respiration. This limits the magnification, especially when compared to what can be achieved in fixed tissue.

3. The authors should note that lipid levels are not simply due to lipogenesis, but could be explained by altered oxidation rates, altered uptake of lipid, decreased lipophagy etc. So the authors should be more circumspect about comments made in line 112.

We included the additional possibilities in the text (see lines 125-126).

Also, for Fig. S2, it should be noted that the lipogenesis genes noted (Fasn, Acly, Acaca, Scd1) are subject to negative feedback control via lipid-controlled regulation of Srebps so one would

expect to see them up-regulated at the transcriptional level in regions where there is less lipid so the comment in lines 117 – 119 needs to be amended.

The text was revised accordingly.

4. The area of most concern pertains to the analysis of mitophagy. Up to figure 5, all data points to there being more mitochondria, smaller mitochondria and increased OCR/ATP generation in PP regions which in turn would suggest decreased mitophagy in the PP region since, as the authors mention, mitophagy preferentially eliminates smaller mitochondria with fused mitochondria spared. Plus, as the authors show, protein expression of Bnip3, Bcl2l13, Gabarapl3 and Gabarap are all increased in PC regions (Fig. 5C). Plus, if you look at Fig. 5B, you see increased mitophagy with Leupeptin treatment most in the PC region while there is minimal mitophagic flux in the PP region, at least based on the histogram in Fig. 5B. It would be good to see the actual images for this data to compare to Figure 5A. It seems as if there may just not be very much mitophagy at all in the liver shown in Figure 5A. Indeed, all of this is done under fed conditions, and liver mitophagy is most induced by fasting of mice. The authors should show mt-mKeima images for livers from fed and fasted mice and plus/minus leupeptin and graph accordingly.

These are great comments which were also raised by reviewer #3. We re-evaluated the data interpretation and agreed that the conclusion from the graph shown in Figure 5C (Figure 5B in original submission), is that there is a higher mitophagy flux in PC, **not** PP hepatocytes. This is supported by new data showing similar trends measuring Bnip3 levels using Western blot (Figure 5D and E), but not LC3 (Figure 5D and F). Per the reviewer's request, images of leupeptin-treated mtKeima mice were added (Figure 5B). We have not included fasted mice in this study because it is beyond the scope of the manuscript. Our goal was to define mechanisms that allow mitochondria that are 300um apart to maintain distinct identities. Since mitochondria turnover helps regulate bioenergetics, we decided to ask whether mitophagy was differentially regulated across the lobule. Indeed, fasting robustly activates hepatic mitophagy which is a topic of active investigation in the lab.

5. Turning to SFig.4 which also addresses mitophagy, the text on line 167 claims that Sfig.4A shows “accumulation of mitochondria in lysosomes” which it really does not – it is a western blot of processed LC3 from whole liver from fed mice plus/minus leupeptin. Images of mitochondria inside lysosomes would indeed be very useful here but that is not what is shown so the text needs correcting or images of “mitochondria in lysosomes” added to the panel. Also if showing LC3 processing, would be best to perform on lysates from PP versus PC, not from whole liver.

The purpose of the Western blot shown in the original Figure S4A was to show the protease inhibitor leupeptin treatment was effective in blocking autophagy *in vivo*. Nevertheless, we see the benefit of showing LC3 levels in PP and PC-sorted cells. The revised Figure 5 now includes a representative blot of LC3 levels in total, PP and PC cells from fed mice +/- leupeptin treatment, and quantification of experiments in 5-6 mice. We revised the text (lines 263-264) to better describe the leupeptin treatment.

We also added a Western blot analysis of Bnip3 (Figure 5D-E). The blots complement the functional experiments done with intravital microscopy of mKeima mice showing that while basal mitophagy flux is regulated in zoned manner with a higher flux in PC hepatocytes (Bnip3 analysis), basal autophagic flux is uniform (LC3 analysis).

6. The images in SFig.4B and graphed in SFig.4C have been mis-interpreted. Again, there is more flux in the PC region since there is more of an increase in mitophagosomes when leupeptin is added, as is clear from the graph Fig.S4C) – leupeptin really does not change the levels of mitophagosomes in the PP region but does increase them in the PC region which argues that there is more mitophagic flux in the PC region and this is also more consistent with the authors other observations that there are more mitochondria, smaller mitochondria and increased OCR and ATP generation in the PP region. And while the PC hepatocyte starts out with fewer mitochondria (Fig.S4C bottom left), that is as likely due to increased mitophagy *in vivo* that has already taken place since mitophagosomes accumulate in the PC hepatocyte when they block mitophagy with Leupeptin (Fig.S4C bottom right) indicating very nicely that there is more mitophagy in PC cells (blue bars in graph in Fig.S4C increase with Leupeptin illustrate this) than

in PP cells where Leupeptin has minimal effect (red bars in graph in Fig.S4C do not change). Overall, the assessment of there being increased mitophagy in PP regions and less mitophagy in PC regions is incorrect and needs to be re-assessed since the data presented do not support this conclusion. To the contrary, the data support the converse.

We appreciate and agree with the reviewers' interpretations of the data. We revised the text accordingly.

To make a more robust claim about rates of mitophagy, the authors should also perform double labeling for mitochondria and lysosomes (like they do in Figure 1 for CD73 and E-cadherin), perhaps CoxIV and Lamp1 co-staining, plus or minus Leupeptin. Importantly, given that mitophagy and autophagy more generally is induced by fasting, the authors should add data examining the effect of fasting on their observations.

This is a great suggestion. We used livers from Dendra2 mice and stained them with LAMP1 in fed or fasted mice, +/- leupeptin. This is a summary of our observations: 1. There was no visible difference in the distribution of LAMP1-positive structures along the PP-PC axis. 2. Fasting and leupeptin had a visible effect on the number and size of LAMP1-positive structures, respectively, but the differences between PP and PC were diminished. 3. We observed mitochondria inside LAMP1 positive structures (arrowheads) in all conditions in both PP and PC. However, PC hepatocytes seemed to have more mitochondria in LAMP1-positive structures, consistent with higher mitophagy flux in PC regions. 4. Colocalization analysis of overlapping pixels did not yield consistent results. We think that this analysis was inaccurate because LAMP1-positive structures overlap extensively with mitochondria in the cytosol many of which are not engulfed in mitophagosomes. Additionally, Dendra2 is sensitive to the lower pH within mitophagosomes causing reduced fluorescence signal and inaccuracies in quantitative assessment of mitophagy. For these reasons, we decided to include the qualitative evaluation of the results of the fed mice only (Figure S7F). The full panel, including the experiment in the fasted mice, is shown below (Response Figure 1).

Response Figure 1

7. The second area of concern is the interpretation that increased phosphorylation of proteins like Mtf1, Mtf2, Bnip3 and Bcl2l13 results in their activation, as mentioned lines 194 to 197 and 295 - 297. The phosphorylation of these proteins could be inhibitory and this is not at all considered. The interpretation and text here needs to be reconsidered.

This is true and the text was revised. We also color-coded the phospho-sites in the table, based on published literature, to reflect if the site was activating (green), inhibiting (purple), or uncharacterized (black).

8. The authors make claims about AMPK and mTOR activation without measuring zonal differences in their activity. The westerns in Figure S6 need to be repeated on sorted PP and PC hepatocytes. In addition, they should examine how zonal activity of AMPK and mTOR changes in response to fasting.

We want to emphasize that the activation and regulation of mTOR and AMPK are complex, and this study does not provide an in-depth analysis of their coordination in space. The main question of this study is “whether mitochondria variations are determined developmentally (by a molecular gradient along the sinusoid) or metabolically (via dynamic nutrient gradients) requires elucidation” (lines 48-50 in the introduction). In other words, is mitochondrial zonation only driven by zonal gene expression or are there any additional mechanisms? Our hypothesis is that phosphorylation provides a rapid and reversible way to dynamically respond to changes in nutrient gradients via nutrient-sensitive kinases. We added the following to clarify this in the revised manuscript:

1. Analysis of the phosphorylated consensus sequences using <https://meme-suite.org/meme/tools/momo> showing distinct sequences in PP and PC hepatocytes (Sup file 4).
2. Analysis of the zoned mitochondrial phospho-peptide using PhosphoSitePlus, shows that many of them are regulated by upstream factors or kinases related to nutrient availability (i.e., refeeding, leptin, insulin, AMPK, mTOR and their regulators);(Sup file 5).
3. Analysis of putative AMPK and mTOR substrates using GPS 5.0 (<https://gps.biocuckoo.cn/>), an online tool for the prediction of kinase-specific phosphorylation sites (Sup file 6). We further analyzed the data and found that approximately 50% of the quantified PP or PC phosphosites are putative phosphorylation sites for AMPK, mTOR, or both (some of the consensus sequences are very similar) (Figure S8D). While AMPK and mTOR had a similar number of putative

phosphosite in PP and PC, there was very little overlap in substrates (Figure S8E). This suggests that while AMPK and mTOR signaling are active across the lobule, they are modulating different signaling pathways, which may affect mitochondria diversity directly or indirectly. We show evidence in support of this idea with two mTOR substrates: S6K and 4E-BP1 (Figure S8F).

4. Correlation plots between PP/PC mitochondrial proteome and Wnt-regulated genes showed moderate correlation (Figure S9D) supporting the idea that other mechanisms may be at play.

To test whether mitochondria variations are also determined metabolically (via dynamic nutrient gradients), we chose to pharmacologically modulate AMPK and mTOR.

Pharmacological manipulation, as opposed to nutritional manipulation (fasting), allows acute change of nutrient-sensitive kinases with limited or no transcriptional effect. This also decouples phosphorylation from expression. We hope this clarifies our rationale and explains why the activation of AMPK and mTOR in the context of fasting was not examined. In addition, pACC1, pS6K, and p4E-BP1 in sorted cells were added to the manuscript in Figures S4B and S8F.

9. Finally, the graph in figure 7B need to be revisited since if you look at the effects of AMPK and mTOR activators/inhibitors, you can see here again that the blue (PC) numbers are the ones that change the most, and again are more consistent with there being more mitophagy in the PC regions not the PP regions, not less.

We thank the reviewer for this observation; we agree that PC mitochondria are highly responsive to the drug treatment and mentioned this observation in the text.

Minor points

1. Please define GS in figure 1D in the figure legend.

GS definition was added.

2. The text in Figure 2C is too small to be legible.

The font size was amended in Figure 2C.

Reviewer #3

This is a very interesting study on mitochondrial heterogeneity within liver tissue. The paper focuses on characterizing mitochondria from two spatially distinct populations of hepatocytes – those in the periportal (PP) region as well as those in the pericentral (PC) region. Previous literature has documented liver zonation, but studies have focused either on gene expression differences or morphological distinctions (such as those by electron microscopy). While suggestive, these data do not fully reflect the heterogeneity seen in liver tissue, particularly at the level of mitochondrial function. This study addresses this gap by performing thorough and impressive characterization of hepatocyte mitochondrial function in isolated organelles, primary cells, and in vivo.

The study begins with a proteomics analysis on spatially sorted hepatocytes, with PC cells expressing the marker CD73 and PP cells expressing the marker E-cadherin. The immunofluorescent imaging of these regions in Figure 1B is particularly striking. The results from the proteomics experiment suggest that hundreds of mitochondrial proteins are ‘zonated,’ or preferentially expressed in either PP or PC hepatocytes relative to the other cell type. This is extremely interesting. Many studies have suggested mitochondrial heterogeneity across tissues, but this, to my knowledge, is amongst the first demonstrations of such striking proteomic diversity across the same cell type in a singly tissue. Based on these data, the authors hypothesize that PP mitochondria will have greater oxidative capacity relative to PC mitochondria, and, using a variety of techniques both in vivo and in isolated cells, show significant differences in the metabolic programs of PP and PC hepatocytes. Specifically, the authors propose that PP mitochondria are better suited for beta oxidation, while PC mitochondria are more specialized for lipid synthesis. The authors continue in their characterization of PP and PC mitochondria, finding that PP mitochondria are morphologically distinct (i.e., they have distinct sizes and shapes) relative to PC mitochondria, and that PP mitochondria display higher rates of mitophagy. Finally, the authors perform phosphoproteomic analysis of PP and PC hepatocytes. The authors identify numerous phosphorylation events on mitochondrial proteins across various pathways, including mitophagy and fusion/fission, that are spatially enriched in one of the two cell types. The

authors then end the manuscript by determining that two key nutrient sensing kinases, AMPK and mTOR, can modulate a subset of the phenotypes described in PP and PC hepatocytes.

Overall, the authors use a wide breadth of experimental techniques to characterize mitochondrial heterogeneity in mammalian liver tissue. In my opinion, this study is highly innovative and provides not only key techniques to the field (e.g., spatial sorting of hepatocytes), but will provide key datasets for many follow up studies.

Despite these strengths, as well as the rigor in many of the experiments performed in this manuscript, there are some aspects of this study that would be strengthened through additional experimentation to support key claims (see below). I include my concerns and potential experimental suggestions to address these concerns below (see 'Major revisions'). I also include some minor points that would make the manuscript easier to read and/or understand (see 'Minor revisions').

We are grateful for the reviewer's appreciation of the study.

Major revisions

Major revision #1: Addressing claims

This paper puts forth a number of claims (text sourced from the paper abstract), most of which are rigorously tested, including:

1. "...PP and PC mitochondria are morphologically and functionally distinct..."
2. "...beta-oxidation and mitophagy were elevated in PP regions..."
3. "...acute pharmacological modulation of nutrient sensing through AMPK and mTOR shifted mitochondrial phenotypes in the PP and PC regions of intact liver."

However, two claims that may be overstated or should be reworded are as follows:

4. "...lipid synthesis was predominant in the PC mitochondria."

- The authors put forward data consistent with lipid synthesis being elevated in PC mitochondria relative to PP mitochondria, but do not directly test this. (See Major Revision 2 for suggestions on addressing this.)

We re-examined the phosphoproteome data and found that phosphorylation of serine 29 on ACC1 is PP zoned (Figure 6H). In addition, we performed WB using an antibody against ACC1 S79 in sorted cells and found higher levels of phosphorylated ACC1 in PP hepatocytes (Figure S4B). Both phosphosites are inhibitory and therefore indicate inhibition of lipogenesis in PP regions.

5. "...mitophagy and lipid synthesis are regulated by phosphorylation in a zoned manner."

- The authors indeed find that proteins involved in mitophagy and lipid synthesis have distinct, zoned phosphorylation patterns (Figures 6G and H), but they do not put forth any evidence to suggest that these events are regulatory. Such a claim would require more data to be put forth regarding the specific function and the dynamic contexts in which these phosphorylation events are altered. The generation of such data is likely beyond the scope of this manuscript, and I suggest that the authors reword this claim rather than perform extensive follow up experimentation on this point.

The statement was revised.

Major revision #2: More rigorous testing of lipid synthesis in zoned hepatocytes

As noted above, the authors claim that lipid synthesis is predominant in PC mitochondria. The data supporting this claim are:

a. significant elevation of lipid droplet (LD) formation in PC hepatocytes (Figure S2A),

- b. elevated citrate synthase (Cs) activity in PC hepatocytes (Figure 3J), and
- c. elevated TG levels in PC hepatocytes (Figure 3K).

The authors put forth this collection of data to suggest that PC hepatocytes are more lipogenic, and the data are consistent with such a model. However, the data are also consistent with other models. For instance, elevated LDs, as well as elevated TGs (which are likely coupled phenotypes) could be derived from an increase in lipid synthesis (as proposed), decreased LD catabolism (including decreased FAO, which they authors indeed show in Figure 3F), or impaired TG/VLDL secretion (reviewed in PMID: 28428634). Thus the increase in LDs/TGs is not sufficient to claim bona fide lipogenesis.

This point was also raised by reviewer #1. We agree that other models may apply and include them in the text (lines 125-126). In addition, we propose inhibition of lipogenesis in PP hepatocytes by AMPK-dependent phosphorylation of ACC1 on S29 (by phosphoproteomics-now included in the revised table in Figure 6H) and S79 (by WB-in Figure S4B).

Second, the authors claim that Cs activity is increased in PC hepatocytes, however no data on Cs total protein expression is presented. Is it possible that Cs is simply overexpressed in PC hepatocytes relative to PP hepatocytes? It is also well known that Cs activity correlates to total mitochondrial content, particularly in liver as recently tested (PMID: 33077793). While these experiments may still prove consistent with the model that Cs activity is elevated due to lipogenesis, the point in bringing up these other studies is that Cs activity could be altered independent of lipogenesis, and thus is an indirect measure at best.

Citrate synthase (CS) protein expression is among the top 25 proteins enriched in PC hepatocytes shown in Figure 2D. The study mentioned by the reviewer (PMID: 33077793) aimed to identify mitochondrial differences across organs using label-free, mitochondrial-targeted nanoLC-MS/MS. Their conclusion is that CS activity does not accurately reflect mitochondrial content across mouse tissues. Although the liver shows a relatively good correlation between MitoCarta protein enrichment and CS activity, the data is based on mitochondria isolated from the whole liver, including non-hepatocytes, and therefore lacks the

cellular specificity and spatial resolution our study provides. Our findings suggest that in hepatocytes, mtDNA content is a better measure of mitochondrial content.

As one of the focal points of the paper is the nutrient regulation of PP versus PC hepatocytes, the authors should perform functional lipogenesis experiments in both PP and PC hepatocytes to demonstrate that lipid synthesis is elevated in PC relative to PP cells. Experimentally, there are many protocols to assay lipogenesis in primary hepatocytes, including PMID: 26382148. The authors could also bolster this claim by assaying LD formation in a fasting/refed condition – one would expect to see increases in LD formation in this assay in PC versus PP hepatocytes in post-prandial conditions (although admittedly this is an indirect assay). The authors could also mine their phosphoproteomics analysis for well described lipogenic PTMs such as Acc phosphorylation (gene symbol Acaca; phospho-S79 – inhibitory and would be expected to be down in PC hepatocytes if lipogenesis is up; this one is also of particular interest because it is regulated by Ampk) or Acly (phospho-S455 – activating and would be expected to be up in PC hepatocytes).

We thank the reviewer for the thoughtful comments and suggestions. However, we do not expect to see an increase in LDs in a fasting/refeeding experiment. During fasting, LDs accumulate in the liver due to increased flux of free fatty acids from adipose tissue (PMID: 32146030). Therefore, LD content is expected to drop in the refed state. Instead, we mined the phosphoproteomics data, as the reviewer suggested, and found that the phosphorylation of ACC1 on S29 indeed was enriched in PP hepatocytes. Both S29 and S79 are well-characterized sites that when phosphorylated by AMPK, inhibit lipogenesis. We revised the table in Figure 6H to include all the significantly zoned phosphoproteins from the screen, including ACC1 S29. We also show the zonation of the S79 phosphorylation by WB (Figure S4B). We also added a supplementary Excel file listing identified phospho-peptides and normalized to proteome (Supplementary Files 2 and 3).

Major revision #3: Further exploring mitophagy phenotypes and proteins

The authors show very interesting data that mitophagy flux increases in PP mitochondria despite the fact that mitophagy proteins such as Bnip3 and Bcl2l13 are enriched in PC mitochondria. The authors propose this is because “basal mitochondrial is higher in the metabolically active PP hepatocytes, which may maintain the levels of the mitophagy-related proteins low.” This could be tested with the leupeptin experiment shown in Figures 5A/B; if mitophagy proteins are kept low because of high flux, they should accumulate in the presence of a lysosomal blocker. The authors should test this, with any means that is easiest for them to detect mitophagy proteins. (Bnip3 or Bcl2l13 would be preferable to test here because they are reported in the dataset, but others would suffice for this line of experimentation as well.)

Thank you for suggesting the experiment, which is now included in Figure 5D and E. The results indicate that Bnip3 levels are lower in the PP regions because it is expressed at lower levels, not because it is being degraded via mitophagy. We revised our interpretation in the manuscript.

Major revision #4: Maximizing impact with proteomic and phosphoproteomic datasets

a. The authors perform TMT-based proteomics and phosphoproteomics as a part of the characterization of PP and PC hepatocytes in this manuscript. Unfortunately, the raw datasets were not provided to the reviewers, so I could not look into the trends with the data. Upon publication, these datasets should be provided as supplementary information so as to provide an accessible resource for readers of this manuscript. Additionally, the raw datasets should be deposited into a database such as PRIDE.

We apologize for not including the supplementary files in the original submission. In the resubmission, we included the following Excel spreadsheets: 1. Supplementary File 1: Differentially expressed proteins and mitochondrial proteome, 2. Supplementary File 2: Raw phosphoproteome dataset and 3. Supplementary File 3: Phosphoproteomics dataset normalized to protein abundance. For easy navigation, within each document, we included tabs of the various subsets including PC, PP, and UZ (unzonated) proteins based on the directionality of the FC value (positive is PC or negative is PP) and p-value ($P < 0.05$). In addition, tabs

including mitochondrial protein subsets are shown. Once the manuscript is accepted for publication, we will deposit the raw dataset to PRIDE.

b. The proteomics and phosphoproteomics data in PP versus PC hepatocytes are visualized appropriately and show interesting trends. However, I believe the relative protein abundance changes could be confirmed for a handful of targets by a second measure. This is not because I do not believe the proteomics data, but rather because proteins quantified by proteomics, particularly with TMT tags, are prone to dynamic range compression due to precursor interference (see PMID: 24211767). It is thus possible that the differences in protein expression between the two cell types are being underestimated. Immunofluorescence similar to what is shown in Figure 1B for a couple of differentially expressed targets would be powerful validation.

Thank you for suggesting this and we agree that further validation of the proteomics results is required. We added a new Figure S2 showing immunofluorescence staining using antibodies against Aldh1b1 and Oat that are enriched in PP and PC mitochondria, respectively.

c. The schematics in Figure 2C and D would be improved with some type of metric for the fold change between the two cell types (if possible. This is a minor comment.)

Thank you for the suggestion. Figures 2C and D show the top 25 enriched proteins in PP or PC mitochondria, if color-coded based on expression level, they will all have a similar shade. We selected to color code the pathway instead, to highlight the variations in potential functions.

d. Some of the phosphosites in Figure 6H seem to be mislabelled. For instance, Bnip3 is annotated as being phosphorylated at "S78" but residue 78 is a Hisidine (Uniprot: O55003); similarly, Mtfr1 is phosphorylated at "S234/235" but these residues are Pro, Gln (Uniprot: Q99MB2) and Mtfr1l is phosphorylated at "T118" but this is an alanine (Uniprot: Q9CWE0). Bcl2l13, Pkraca, and Apex1 have correct residues at their annotations, but that is as far as I validated. The authors should recheck this; this could also be alleviated by including the identified peptide within a supplemental table (as pointed out in 3a).

We thank the reviewer for catching these errors. We have reviewed the data in Figure 6H and made the following corrections:

Bnip3 is phosphorylated at S79 (not S78)

Mrfr1l is phosphorylated at S234/235 (not T118)

Mrf1 is phosphorylated at T118 (not S234/235)

The peptide sequences can be found in Sup File 2.

e. The authors perform functional experiments with AMPK and mTOR activators/inhibitors and find meaningful changes between PP and PC hepatocytes, but they do not tie this back into the phosphoproteomics analysis. Are there phosphosites that lie within AMPK or mTOR consensus motifs? If so, does this suggest direct regulation of these substrates, or do the authors propose that this is indirect regulation? Such points should be brought up in the discussion (beyond the single example of AMPK-mediated regulation of Mtfr1l. Also, does this citation characterize the same Mtfr1l phosphosite identified in this paper? If so, this should be highlighted.).

This is an excellent point. During the analysis of the zonated mitochondrial phospho-peptide using PhosphoSitePlus, we noticed that many of them are regulated by upstream factors or kinases related to nutrient availability (i.e., refeeding, leptin, insulin, AMPK, mTOR and their regulators). We included the extended table with the resubmission (Sup file 5). We then examined the phosphorylated consensus sequences using <https://meme-suite.org/meme/tools/momo> and found enrichment of distinct sequences in PP and PC hepatocytes (Sup file 2).

Next, we used GPS 5.0 (<https://gps.biocuckoo.cn/>), an online tool for the prediction of kinase-specific phosphorylation sites, to examine whether AMPK and mTOR signaling showed spatial bias. We found that approximately 50% of the quantified phosphosites in PP or PC are putative phosphorylation sites for AMPK, mTOR, or both (some of the consensus sequences are very similar) (Figure S8D-E). While AMPK and mTOR had a putative phosphosite on a similar number of substrates in PP and PC, there was very little overlap (Figure S8F). This suggests that while AMPK and mTOR signaling are active across the lobule, they are activating different signaling pathways, which may affect mitochondria diversity directly or indirectly. To date, hepatocyte

zonation was shown to be driven by differential gene expression, however, this link between mitochondrial zoned phosphoproteome and nutrient signaling raised the intriguing possibility that zoned phosphorylation plays a role in mitochondrial heterogeneity across the lobule. The fact that there was only a moderate correlation between Wnt-regulated genes and zoned mitochondrial protein expression (Figure 7A) supported the idea that other mechanisms may be at play. We hope this clarifies the link between the phosphoproteome data with the experiments modulating AMPK and mTOR signaling. We made sure to include this in the text. Finally, yes, the AMPK-mediated phosphorylation of Mtf1l we identified on S235 (Figure 6H) is the site described in the cited paper. Mtf1l S235 (S238 in humans) is conserved among vertebrates. Phosphorylation of Mtf1l on Ser103 or Ser238 by AMPK is required for Mtf1l function of regulating mitochondrial morphology by stress-induced mitochondrial fragmentation (PMID: 36367943).

f. Pdha1 phosphorylation is elevated in PC hepatocytes (Figure 6H), consistent with decreased flux through pyruvate dehydrogenase. This is somewhat at odds with data presented in Figure 3H showing that PC hepatocytes are more susceptible to loss of ATP in the presence of UK5099, an inhibitor of pyruvate uptake. The authors should comment on this.

This is true, the inhibitory phosphorylation of Pdha1 is higher pericentrally, which seemingly contradicts the Seahorse data suggesting PC hepatocytes rely on pyruvate for ATP production. However, Pdha1 is also PC zoned, meaning its relative expression levels are higher in PC hepatocytes which could mean that there is more active enzyme (not phosphorylated) in PC hepatocytes than PP.

Minor revisions:

1. Cellular ATP is presented as a normalized percentage in Figure 3I; this data would be stronger presented as absolute values.

Thank you for the suggestion. We have replaced the graph in Fig 3I with the data expressed as ATP measures (nM) and normalized to protein levels.

2. The authors note that PP hepatocytes are smaller than PC hepatocytes, and that modulation of mTOR can change some of these effects. mTOR is well known to control cell size (PMID: 12080086) – do the authors see key targets of mTOR such as S6 differentially phosphorylated in PP versus PC hepatocytes that could contribute to such phenotypes?

We want to emphasize that the activation and regulation of mTOR and AMPK are complex, and our study does not provide an in-depth analysis of their spatial coordination. As mentioned earlier, we used bioinformatic tools to identify AMPK and mTOR putative consensus sequences and show that both AMPK and mTOR are active throughout the lobule but activate distinct substrates in different parts of the lobule (Figure S8 D-F). For example, we show the differential phosphorylation levels of two mTOR substrates, S6K and 4E-BP1 in sorted cells (Figure S8E). We also provide the full list of AMPK and mTOR putative substrates derived from the phosphoproteomics dataset using GPS 5.0 (Figure S8F and Sup file 6). The higher pS6K levels in PC hepatocytes are consistent with our observations that PC cells are larger.

REVIEWER COMMENTS

Reviewer #1 (Remarks to the Author):

This is a revised version of a manuscript making several important observations about mitochondrial protein expression and mitochondrial function across the liver lobule in response to nutrient availability. This work critically extends analysis of liver zonation beyond RNA expression which has till now been the main basis on which zonation has been characterized. The authors have responded very effectively to the previous critique and overall the manuscript is convincing and makes a substantive contribution to the field.

A couple of minor concerns remain mainly surrounding the data and conclusions reached about AMPK and mTOR activity in liver zones. Specifically, the western in Fig.S8F is contradictory to some extent in showing opposite effects of zonation of pS6K1 and p4EBP1. Ideally, the authors should examine additional mTOR targets, such as pSrebp1 and Lipin1, and also western blot for AMPK targets, for example pAMPK, pRaptor to permit them to make the conclusions they do about these two critical kinases. Related to this, the effects described on page 9, lines 236 – 238 are not borne out by the data shown in figure 7B, and in fact the wrong panel appears to be cited here (Fig.7C, line 238 when it should be 7B). Finally the effects of AMPK inhibition and mTOR activation on mitochondria as described on page 10, lines 243 – 248 are again not consistent with the figure (Fig.7C) where the effects of the drugs appear to be greater on mitochondrial mass (lower with AMPK inhibition and greater with mTOR activation – and consistent with known roles of these kinases on autophagy) than on mitochondrial shape, and as supported by the unconvincing shape data in Fig.7D. Finally, the authors may want to show a western for GABARAP in Fig.5D, rather than LC3A/B given that they show GABARAP to be most significantly altered in PP vs PC hepatocytes in Fig.S7D.

Minor edit: Page 9, line 212 – remove second “for”.

Reviewer #2 (Remarks to the Author):

This is a highly interesting study characterizing the differences in composition and function of hepatocytes across different regions of the liver. The revision is well done, and the new data substantially contribute to the study. The addition of the FIB-SEM data is particularly striking. This manuscript will be of high interest and value to many scientists, and it should be accepted in its current form.

I did find a handful of small errors for the authors to note, but do not think these should prevent acceptance for publication.

1. In Figure 5D, the Western blots are missing molecular weight markers.
2. In Figure 5B, the inlays/zoom ins are missing scale bars.
3. In line 298 (bottom of page 11), the authors annotate Pdha1 as a mitochondrial kinase; this is the E1 subunit of PDH and is not a kinase. The authors could amend this statement by simply deleting Pdha1 from this sentence.

Manuscript NCOMMS-23-17889A

A spatial map of hepatic mitochondria uncovers functional heterogeneity shaped by nutrient-sensing signaling

We thank the reviewers for the additional comments and have addressed them in the point-by-point response below.

REVIEWER COMMENTS

Reviewer #1 (Remarks to the Author):

This is a revised version of a manuscript making several important observations about mitochondrial protein expression and mitochondrial function across the liver lobule in response to nutrient availability. This work critically extends analysis of liver zonation beyond RNA expression which has till now been the main basis on which zonation has been characterized. The authors have responded very effectively to the previous critique and overall the manuscript is convincing and makes a substantive contribution to the field.

We greatly appreciate the reviewer's acknowledgment.

A couple of minor concerns remain mainly surrounding the data and conclusions reached about AMPK and mTOR activity in liver zones. Specifically, the western in Fig.S8F is contradictory to some extent in showing opposite effects of zonation of pS6K1 and p4EBP1. Ideally, the authors should examine additional mTOR targets, such as pSrebp1 and Lipin1, and also western blot for AMPK targets, for example pAMPK, pRaptor to permit them to make the conclusions they do about these two critical kinases.

The WBs in Figure S8F demonstrate that mTOR is phosphorylating different substrates and potentially activating different signaling pathways in PP and PC hepatocytes. Although 4E-BP inhibition and S6K activation are both downstream of mTORC1 activation, and both promote protein synthesis, previous studies suggested that S6K and 4E-BP differentially control cell growth and proliferation. S6K controls cell size but not cell cycle progression (PMID: 15723049), whereas 4E-BP controls cell proliferation but not cell size (PMID: 20508131). PP hepatocytes produce albumin and clotting factors posing a high energetic demand for serum protein synthesis (PMID: 28166538). This is consistent with highly bioenergetic mitochondria and higher phosphorylation of 4EBP1 that allows protein synthesis. On the other hand, PC hepatocytes are larger (Figure S9C) which is consistent with higher levels of pS6K1.

The goal of the study was to examine how the nutrient gradient along the PP-PC axis contributes to the zonation of the mitochondria phosphoproteome. Since this axis is roughly 300µm long and is impossible to manipulate the nutrient concentrations experimentally, we

chose to acutely modulate hepatocytes' perception of nutrient availability instead. We focused on mTOR and AMPK because they were identified as upstream regulators of many of the phosphosites identified in the screen. Furthermore, the pharmacological tools to modulate these pathways are well established. The current study does not aim to investigate AMPK and mTOR zonation, which would require genetic manipulation and is beyond the scope of this study. Many of the targets the reviewer suggested were identified in the proteome and phosphoproteome screen (supplementary files 1-3) and interested readers will be able to use that in their studies.

Related to this, the effects described on page 9, lines 236 – 238 are not borne out by the data shown in figure 7B, and in fact the wrong panel appears to be cited here (Fig.7C, line 238 when it should be 7B).

We appreciate the reviewer noted the wrong panels on page 9 were cited in the text and have corrected this.

Finally the effects of AMPK inhibition and mTOR activation on mitochondria as described on page 10, lines 243 – 248 are again not consistent with the figure (Fig.7C) where the effects of the drugs appear to be greater on mitochondrial mass (lower with AMPK inhibition and greater with mTOR activation – and consistent with known roles of these kinases on autophagy) than on mitochondrial shape, and as supported by the unconvincing shape data in Fig.7D.

We appreciate the comment. The images shown in Fig 7C are confocal images of a tissue plane aimed to demonstrate the striking effect of the drugs on mitochondrial morphology. We see how the thin section can create the wrong impression and therefore added z-stack volumes in Fig S9E. We hope that despite the density of the mitochondria, these volumes demonstrate why we think activation of AMPK or mTOR affects primarily mitochondrial morphology. We also observed that AICAR-treated PP hepatocytes had higher mtDendra2 intensity, which may be linked to AMPK's role in mitochondrial biogenesis. Nevertheless, we refrained from making any statements regarding this because AMPK and mTOR signaling, like the balance between mitochondrial degradation and biogenesis, are complex processes. Gain and loss of function studies will be necessary to dissect these pathways, which are beyond the scope of the current study.

Finally, the authors may want to show a western for GABARAP in Fig.5D, rather than LC3A/B given that they show GABARAP to be most significantly altered in PP vs PC hepatocytes in Fig.S7D.

We are grateful for the suggestion and may try it in the future. For the conclusions made in the current study, we feel the blot for LC3A/B is sufficient.

Minor edit: Page 9, line 212 – remove second “for”.

Thank you, this was removed.

Reviewer #2 (Remarks to the Author):

This is a highly interesting study characterizing the differences in composition and function of

hepatocytes across different regions of the liver. The revision is well done, and the new data substantially contribute to the study. The addition of the FIB-SEM data is particularly striking. This manuscript will be of high interest and value to many scientists, and it should be accepted in its current form.

We greatly appreciate the reviewer's acknowledgment.

I did find a handful of small errors for the authors to note, but do not think these should prevent acceptance for publication.

1. In Figure 5D, the Western blots are missing molecular weight markers.

Figure 5D was revised to include molecular weight markers.

2. In Figure 5B, the inlays/zoom ins are missing scale bars.

A scale bar was added to the insets in Figure 5B.

3. In line 298 (bottom of page 11), the authors annotate Pdha1 as a mitochondrial kinase; this is the E1 subunit of PDH and is not a kinase. The authors could amend this statement by simply deleting Pdha1 from this sentence.

This was corrected in the text as the reviewer suggested.

REVIEWERS' COMMENTS

Reviewer #1 (Remarks to the Author):

The authors should include their statement, or something similar to the following "Although 4E-BP inhibition and S6K activation are both downstream of mTORC1 activation, and both promote protein synthesis, previous studies suggested that S6K and 4E-BP differentially control cell growth and proliferation. S6K controls cell size but not cell cycle progression (PMID: 15723049), whereas 4E-BP controls cell proliferation but not cell size (PMID: 20508131). PP hepatocytes produce albumin and clotting factors posing a high energetic demand for serum protein synthesis (PMID: 28166538). This is consistent with highly bioenergetic mitochondria and higher phosphorylation of 4EBP1 that allows protein synthesis. On the other hand, PC hepatocytes are larger (Figure S9C) which is consistent with higher levels of pS6K1." in the discussion section so that readers not familiar with this could appreciate such dichotomies.

Manuscript NCOMMS-23-17889C

A spatial map of hepatic mitochondria uncovers functional heterogeneity shaped by nutrient-sensing signaling

We thank the reviewers for the additional comments and have addressed them in the point-by-point response below.

While creating the Data Source file, we discovered that in experiments shown in Figure 7B and Sup Fig 9C, errors in Excel data pasting were made. The values were corrected in Data Source File, and Figures. Although there are some changes to the drug treatment effect, this had no impact on the results reported in the manuscript or the conclusions.

In addition, we incorporated the graphical abstract as a new Figure 8 to summarize the findings of the study.

REVIEWER COMMENTS

Reviewer #1 (Remarks to the Author):

The authors should include their statement, or something similar to the following "Although 4E-BP inhibition and S6K activation are both downstream of mTORC1 activation, and both promote protein synthesis, previous studies suggested that S6K and 4E-BP differentially control cell growth and proliferation. S6K controls cell size but not cell cycle progression (PMID: 15723049), whereas 4E-BP controls cell proliferation but not cell size (PMID: 20508131). PP hepatocytes produce albumin and clotting factors posing a high energetic demand for serum protein synthesis (PMID: 28166538). This is consistent with highly bioenergetic mitochondria and higher phosphorylation of 4EBP1 that allows protein synthesis. On the other hand, PC hepatocytes are larger (Figure S9C) which is consistent with higher levels of pS6K1." in the discussion section so that readers not familiar with this could appreciate such dichotomies.

We added this to the discussion.